



Manuscript for Atmos. Chem. Phys.

**Space-time variability of ambient PM$_{2.5}$ diurnal pattern over India**
**from 18-years (2000-2017) of MERRA-2 reanalysis data**

**[1]Kunal Bali*, [1,2]Sagnik Dey, [1]Dilip Ganguly and [3,4]Kirk R. Smith**
[1]Centre for Atmospheric Sciences, Indian Institute of Technology Delhi, Hauz Khas,
New Delhi, India
[2]Centre of Excellence for Research on Clean Air, Indian Institute of Technology
Delhi, Hauz Khas, New Delhi, India
[3]School of Public Health, University of California, Berkeley, USA
[4]Collaborative Clean Air Policy Center, New Delhi, India
*Corresponding author: kunal.bali9@gmail.com











Manuscript for Atmos. Chem. Phys.

**Abstract**
Estimating ambient $PM_{2.5}$ (fine particulate matter) concentrations in India over many
years is challenging because spatial coverage of ground-based monitoring, while
better recently, is still inadequate and satellite-based assessment lacks temporal
continuity. Here we analyze MERRA-2 reanalysis aerosol products to estimate $PM_{2.5}$
at hourly scale to fill the space-time sampling gap. MERRA-2 $PM_{2.5}$ are calibrated
and validated ($r = 0.94$, slope of the regression $= 0.99$) against coincident in-situ
measurements. We present the first space-time variability of ambient $PM_{2.5}$ diurnal
pattern in India for an 18-year (2000-2017) period. Diurnal amplitude is found to be
quite large (>30 $\mu g\ m^{-3}$) in the Indo-Gangetic Basin (IGB) and western arid regions of
India. $PM_{2.5}$ is found to decrease over the western dust source region and increase
over the Himalayan foothills and parts of IGB and central India primarily in the
morning and evening hours. This increasing trend at an annual scale is primarily
governed by a large increase in concentration during Oct-Feb that can be attributed to
a combination of the rise in emission and declining boundary layer height. Our results
suggest that the satellite-based concentration estimates (typically representative of late
morning to early afternoon hours) are lower (magnitude depends on the place and
season) than the 24-hour average concentration in most parts of India. In the future,
the integration of reanalysis data in concentration modeling may assist in reducing the
uncertainty in estimates of air pollution concentration patterns in India and elsewhere.







Manuscript for Atmos. Chem. Phys.

**1. Introduction**

Ambient PM$_{2.5}$ (fine particulate matter smaller than 2.5 $\mu$m) concentration is a leading risk factor of global burden of disease (Cohen et al., 2017). In India almost the entire population is exposed to annual PM$_{2.5}$ exceeding World Health Organization (WHO) annual air quality guideline of 10 $\mu$g m$^{-3}$. Even the annual national ambient air quality standard (NAAQS) of 40 $\mu$g m$^{-3}$ was apparently exceeded by about 77% of the population in 2017, resulting in 0.67 (95% uncertainty interval [UI] 0.55-0.79) million premature death and average loss of 0.9 (0.8-1.1) years of life expectancy (Balakrishnan et al., 2017). Ambient PM$_{2.5}$ concentration is further projected to increase in future as India is expected to develop rapidly (Chowdhury et al., 2018; GBD MAPS Working Group, 2018). This calls for urgent implementation of an efficient air quality management plan in India to achieve a sustainable environment, for which a major step is development of a robust air quality monitoring system.

The biggest challenge in monitoring ambient PM$_{2.5}$ in India is lack of adequate ground-based measurements across the country. PM$_{2.5}$ monitoring started in India by the Central Pollution Control Board (CPCB) in 2008-2009. Though the network has expanded since then, still in 2018, India has just 1 monitor for ~7 million population (Martin et al., 2019). The existing monitor density is much lower than that in China (1.2 monitors per million people) where ambient PM$_{2.5}$ concentration was in the same range of India few years ago but started decreasing in the recent years (Zhao et al., 2018). To address this limitation in adequate spatial coverage of PM$_{2.5}$ monitoring in India (and also in many other developing countries), a methodology evolved first to infer PM$_{2.5}$ from satellite-retrieved aerosol optical depth (AOD) following a regression-based approach (Hoff et al., 2009). Later spatially and temporally varying scaling factors derived from chemical transport models were applied (van Donkelaar



Manuscript for Atmos. Chem. Phys.

et al., 2010, 2016; Brauer et al., 2015). This satellite-based approach has been adopted
to generate district-level $PM_{2.5}$ concentration data for India (Dey et al., 2012;
Chowdhury et al., 2016). Though satellite data provide adequate spatial coverage,
they are temporally discontinuous as the passive sensors flying onboard polar
satellites (e.g. Terra and Aqua) can only retrieve AOD during daytime and when they
are overhead. Furthermore, satellite AOD retrieval depends on availability of cloud-
free condition and hence aerosol climatology during the monsoon (June-September)
season in India is biased towards the dry days (Dey and Di Girolamo, 2010).

So far, three approaches have been adopted to address the temporal gap in

sampling. First, AOD data from geostationary satellites allowed continuous $PM_{2.5}$
retrieval over a particular region throughout the day as long as the sunlight is
available for AOD retrieval in cloud-free condition (e.g. Chudnovsky et al., 2012;
Lennartson et al., 2018). Even then, it is not possible to retrieve $PM_{2.5}$ estimates after
sunset. Secondly, integration of $PM_{2.5}$ data from ground-based and satellite
measurements in a Bayesian framework allowed filling the spatial and temporal gap
in the concentration data (Shaddick et al., 2016, 2017). Thirdly, chemical transport
modeling (CTM) has the capability of simulating $PM_{2.5}$ at hourly scale (Michael et al.,
2013), but accuracy of model-simulated $PM_{2.5}$ depends on the model physics,
configurations and representativeness of the emission inventory. In this work, we
propose another approach for estimating ambient $PM_{2.5}$ concentration using reanalysis
aerosol product. Reanalysis products generated by joint assimilation of meteorological
and aerosol observations into global assimilation system take advantages of the best
features of both observations and models (Randles et al., 2017) so that the space-time
continuity in data is maintained with less uncertainty (Bocquet et al., 2015). We
analyze 18-year (2000-2017) of ambient $PM_{2.5}$ concentration at hourly scale and



Manuscript for Atmos. Chem. Phys.

report their climatology, trends and the diurnal amplitude at seasonal scale in India for
the first time.
**2. Approach and Methodology**
We analyze MERRA-2 (Modern-Era Retrospective analysis for Research and
Application) aerosol reanalysis data for this work, which are generated by GEOS-5
atmospheric model at $0.5° \times 0.625°$ horizontal resolution (Gelaro et al., 2017). A
radiatively coupled version of the GOCART (Randles et al., 2016) model that
considers the sources, sinks and chemistry of 15 externally mixed aerosol species -
dust in 5 size bins, sea-salt in 5 size bins, hydrophilic and hydrophobic organic carbon
(OC) and black carbon (BC), and sulfate is used to generate MERRA-2 aerosol
products (Randles et al., 2017). While emissions and transportations of dust and sea-
salt are wind-driven, anthropogenic (EDGARv4.2) and biogenic sulfate (AeroCom
Phase II) and carbonaceous aerosols (scaled RETROv2) are redistributed by winds
after they are emitted (Lana et al., 2011). MERRA-2 aerosol analysis assimilates
quality-controlled AOD at 550 nm from three different sensors (MISR, MODIS Terra
and MODIS Aqua, AVHRR and AERONET). More details of the MERRA-2 aerosol
algorithm are provided in (Randles et al., 2016), while extensive validation of the
product is discussed in Buchard et al (2017). One of the MERRA-2 reanalysis aerosol
products is hourly concentration of individual aerosol species smaller than 2.5 $\mu$m. To
obtain total PM$_{2.5}$, we simply add up dust and sea-salt in size bins smaller than 2.5
$\mu$m, hydrophilic and hydrophobic OC, BC and sulfate (assuming the entire load is
within PM$_{2.5}$). We note that BC, dust and sea-salt are primary particles and sulfate and
OC are secondary aerosols.
We calibrate hourly MERRA-2 PM$_{2.5}$ with coincident PM$_{2.5}$ data from 80 CPCB
sites across the country for the period 2009-2017, as CPCB data are available only for





Manuscript for Atmos. Chem. Phys.


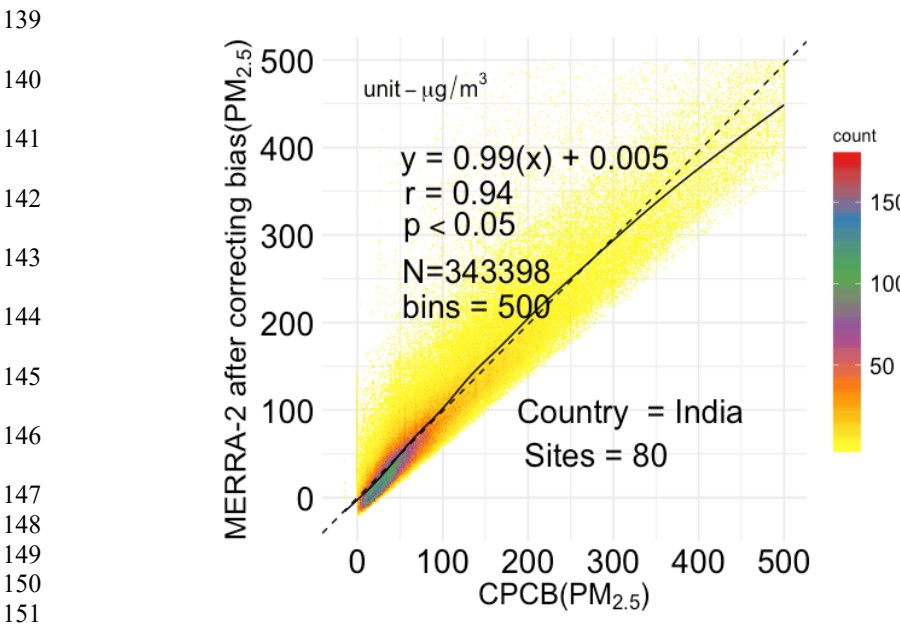

**Figure 1.** Scatter plot between calibrated MERRA-2 PM$_{2.5}$ and in-situ PM$_{2.5}$ data.
Spatial and temporal matching is done by averaging data from all ground-based
monitoring sites falling within a single MERRA-2 grid for 1-hr duration.

this time period. We use half of the data to calibrate and the rest to validate. For
calibration, all CPCB sites within a MERRA-2 grid (0.5° ×0.625°) are averaged. We
identify a low bias in MERRA-2 PM$_{2.5}$ [ΔPM$_{2.5}$ = CPCB-MERRA-2], which increases
linearly with an increase in CPCB-PM$_{2.5}$. We compute the calibration factor at hourly
scale based on the regression for each 1 $\mu$g m$^{-3}$ PM$_{2.5}$ bin, using which we correct the
bias in MERRA-2 PM$_{2.5}$. For the calculation of the bias correction, we followed the
steps give below:
For in-situ 80 sites:
• cpcb (x) Vs merra-2 (y)      ; y = 0.228(x) +23.675
• cpcb (x) Vs difference (y)  ; y = 0.772(x) -23.675  ; [difference = CPCB -
MERRA2]
• calibration factor           = 0.772(cpcb values) + (- 23.675)
• bias corrected merra-2 (BCM) = calibration factor + merra-2
• cpcb Vs BCM ; y = 0.99(x) + 0.005



Manuscript for Atmos. Chem. Phys.

For Indian grid
First
• merra-2 (x) Vs difference (y) ; y = 0.293(x) + 32.693 [for 80 sites]
then calculated the calibration factor for each grid via:
• calibration factor = 0.293(merra grid values) + 32.693
• bias corrected merra-2 (BCM) = calibration factor +merra-2 (grid_values)

The bias-corrected MERRA-2 $PM_{2.5}$ shows statistically significant correlation
(r=0.94, p<0.05) with CPCB ground-based $PM_{2.5}$ with the slope and intercept close to
the ideal values (Figure 1). The regression also reveals that the bias-corrected
MERRA-2 $PM_{2.5}$ data are uniformly spread along 1:1 line below 100 $\mu g$ m$^{-3}$ where
most of the data points lie as well as at high $PM_{2.5}$ values (>100 to 500 $\mu g$ m$^{-3}$). This
justifies the utility of calibrated reanalysis data in examining diurnal pattern of
ambient $PM_{2.5}$ concentration for the entire country, even where no ground-based
monitors are available to calibrate MERRA-2 $PM_{2.5}$. Climatology for hourly ambient
MERRA-2 $PM_{2.5}$ concentration (hereafter we only refer to calibrated $PM_{2.5}$) is
estimated by averaging $PM_{2.5}$ for that particular hour in each day over the appropriate
timescales. Trends are computed using linear regression over the 18-year period.
Diurnal amplitude is estimated as the difference between maximum and minimum
$PM_{2.5}$ in each 24-hour cycle, which is then averaged over the desired timescale to
estimate the climatology. We identify the time of the maximum and minimum $PM_{2.5}$
within 24-hour duration in each season to understand the diurnal pattern of ambient
$PM_{2.5}$ concentration in India over the last 18 years. We also analyze planetary
boundary layer (PBL) height and precipitation rate (PR) at the same hourly scale from
MERRA-2 reanalysis data to understand their influence in modulating the observed
diurnal pattern in ambient $PM_{2.5}$ concentration. Since MERRA-2 aerosol reanalysis
data provide information on speciation, we also examine the observed trends in $PM_{2.5}$



Manuscript for Atmos. Chem. Phys.

in view of changing patterns of these individual components to interpret the
dynamics. In addition to this, diurnal variation of MERRA-2 $PM_{2.5}$ also validated with
CPCB $PM_{2.5}$ (supplementary information, SI, Figure R1) that further shows a strong
correlation (r = 0.8).
**3. Results**
**3.1 Diurnal amplitude of ambient $PM_{2.5}$ concentration in India**
Ambient $PM_{2.5}$ measurements using ground-based data (e.g. Apte et al., 2011; Goel
et al., 2015) suggest a large variation in concentration within a day. However, limited
(spatially) in-situ data hinder development of a regional picture. Satellite-based
concentration estimates (van Donkelaar et al., 2010; Dey et al., 2012; Apte et al.,
2015) assume that the concentration during satellite crossing time is representative of
the 24-hour period. Hence, with large diurnal amplitude in $PM_{2.5}$ concentration, such
estimates may not be a good representative. Therefore, first we report the spatial
patterns of diurnal amplitude in ambient $PM_{2.5}$ concentration in India (Figure 2).
During the post-monsoon and winter seasons (October-February), diurnal ambient
$PM_{2.5}$ concentration varies by >30 $\mu g\ m^{-3}$ in the entire Indo-Gangetic Basin (IGB) and
in the western arid region, which has been identified as a major dust source (Dey et
al., 2012). In January-February, similarly large diurnal amplitude is observed in some
parts of south India, where the amplitude decreases slightly (~20 $\mu g\ m^{-3}$) in October-
December. In the rest of India, ambient $PM_{2.5}$ diurnal concentration varies by a
smaller magnitude. In summer, diurnal amplitude decreases in most parts of India
except the arid regions where it enhances further to >50 $\mu g\ m^{-3}$. Overall, least diurnal
amplitude is observed in the monsoon (June-September) season.
To understand the driving factors of the observed diurnal amplitude, we analyze
test correlation between $PM_{2.5}$ and PBL height and PR (Figure 3). As expected, PBL




Manuscript for Atmos. Chem. Phys.

height and PR show strong negative correlation with PM$_{2.5}$ concentration because
deeper PBL facilitates pollution dispersion (Nakoudi et al., 2018) and large PR leads
to washout of pollution. The arid region in the western India barely receives rain large
enough to influence PM$_{2.5}$ diurnal pattern, which can be attributed to the observed
poor correlation ($r = \sim$-0.3). In the high altitude regions (e.g. in the lower Himalayan
belt and Western Ghats), the pollution is lifted up from the valley beneath as the PBL
expands during daytime and therefore PBL height and PM$_{2.5}$ concentration are
moderately correlated at hourly scale (Srivastava et al., 2012).







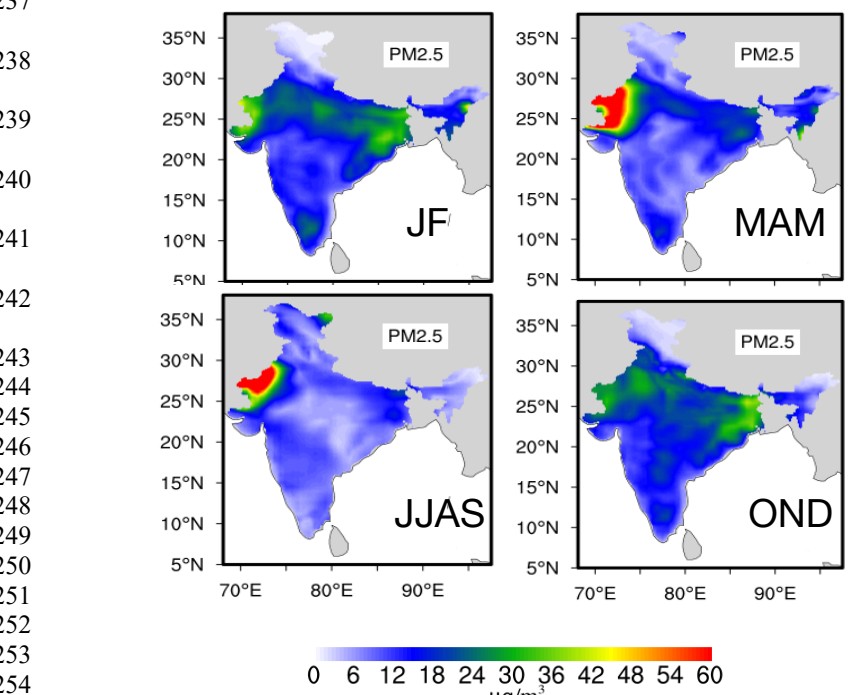

**Figure 2.**    Mean (over 18-years) seasonal diurnal amplitude in ambient PM$_{2.5}$
concentration in India.




Manuscript for Atmos. Chem. Phys.









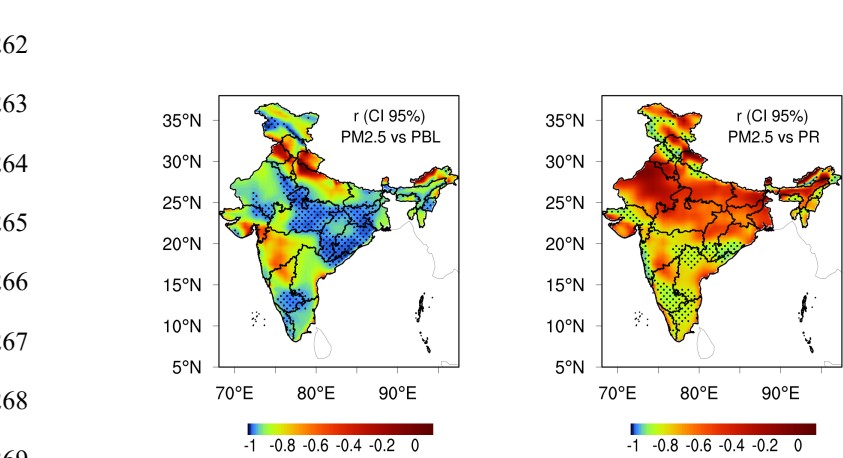


**Figure 3**. Spatial distribution of correlation coefficients between ambient PM$_{2.5}$
concentration and (a) planetary boundary layer (PBL) and (b) precipitation rate (PR)
with the hatched regions showing 95% CI.
We next examine the seasonal shifts of the timings (in Indian standard time, IST)
when PM$_{2.5}$ concentration is observed to be the highest and lowest within a 24-hour
period (Figure 4). In a large part of the country, ambient PM$_{2.5}$ concentration peaks
around early morning hours (6-8 IST) during October-February. In the eastern part of
the IGB and northeastern India, the peak hours are around midnight during these
months, while in the northern hilly regions peak hours are in the late afternoon-early
evening after the PBL fully evolves. The 'PBL expansion' effect is not so prominent
along the Western and Eastern Ghat mountain ranges in these months, but can be seen
in the summer months. The lowest concentration is found during 13-15 IST in the
post-monsoon season that gradually shifts to evening hours (16-20 IST) in most parts
of the country during the winter. Similar timing (late afternoon) in the lowest PM$_{2.5}$
concentration is observed in the central India, parts of north and south India during
Mar-May that further shrinks to only a part of central and western India during the
monsoon season. During these months, the lowest PM$_{2.5}$ concentration is observed





Manuscript for Atmos. Chem. Phys.

during 10-12 IST in the rest of the country.

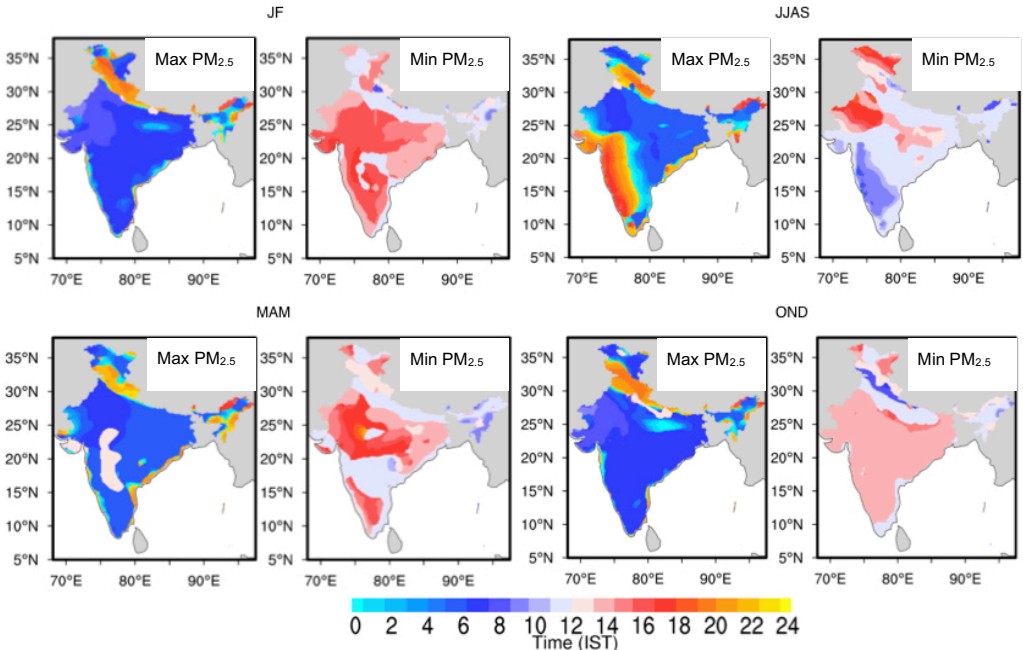

**Figure 4.** Seasonal shifts in the timings of the maximum and minimum ambient PM$_{2.5}$
concentration in India.

**3.2 Space-time variability of ambient PM$_{2.5}$ concentration at hourly scale**
Spatial distribution of annual ambient PM$_{2.5}$ concentration at hourly scale in Figure
5 (statistics at seasonal scale and the corresponding anomaly relative to 24-hr average
are shown in supplementary information, SI) reveals large (>12 $\mu$g m$^{-3}$) positive
anomaly relative to 24-hr average during night and early morning hours over the arid
regions in the west, IGB and parts of peninsular India. Key notable features are as
follows. North (high PM$_{2.5}$)-south (low PM$_{2.5}$) spatial gradient in PM$_{2.5}$ is maintained
throughout 24-hour duration. Even during the noontime and afternoon hours when the
ambient PM$_{2.5}$ concentration is at its minimum in most parts of India, it remains
higher than the NAAQS in the IGB and western arid region. As discussed in the



Manuscript for Atmos. Chem. Phys.

previous subsection, these two regions have the largest diurnal amplitude in $PM_{2.5}$
concentration throughout the year. In the Peninsular India, diurnal variation is less
prominent at the annual scale though the magnitude varies from season to season (see
SI). During January-February, ambient $PM_{2.5}$ concentration in India at 15-16 IST is
best representative of the 24-hr average. $PM_{2.5}$ concentration is higher than the 24-
hour average by >10 $\mu$g m$^{-3}$ in most parts of the country during the late evening to
early morning hours with higher values in the IGB from 22:00 to 04:00 IST. During
the morning hours, $PM_{2.5}$ concentration decreases as the PBL expands. Similar diurnal
pattern is observed during March-May and October-December but with larger diurnal
amplitude. In June-September, diurnal variation is only prominent in the western arid
region.

We further examine the annual trends of ambient $PM_{2.5}$ for every hour (Figure 6).

Positive trend over the 18 years is observed in most parts of India with values
exceeding 1 $\mu$g m$^{-3}$ per year in the Himalayan foothills, northeastern India, eastern
IGB and parts of western IGB. In most part of the Indian regions, the observed
positive trend at annual scale (>2 $\mu$g m$^{-3}$ per year) is largely governed by a massive
increase during October-February (see SI for the trends at seasonal scale). Since
MERRA-2 data are available for individual species, we examine their trends
separately.

Earlier studies (e.g. Verma et al., 2012) showed that sulfate aerosols account for

29% of AOD over the Indian region. We also observe a larger positive trend over 18
years of sulfate aerosols (>0.6 $\mu$g m$^{-3}$ per year) as compared to other aerosol species
(see Figures S17-S19 in SI) throughout the 24-hour period over the IGP and eastern
India (Figure 7). Unfortunately, no available emission inventory captures
anthropogenic emissions at hourly scale. Along with the traditional anthropogenic





Manuscript for Atmos. Chem. Phys.

pollution sources (*e.g.* household emission, vehicles, industries, construction activities
etc.), open burning of agricultural crop-waste and brick kilns add to the emission in
the dry season. Trend analysis of PBL height (see SI) suggests that PBL height is
becoming shallower over the years in most parts of India during the dry season with
higher rate in the late afternoon to early morning hours. Usually the anthropogenic
activities peak during the morning hours and the emissions from certain sectors (e.g.
vehicles, industries, construction etc.) are expected to subside in the late evening-early
morning hours. Combined effect is nearly similar trend in PM$_{2.5}$ concentration in the
dry season throughout 24-hour period.

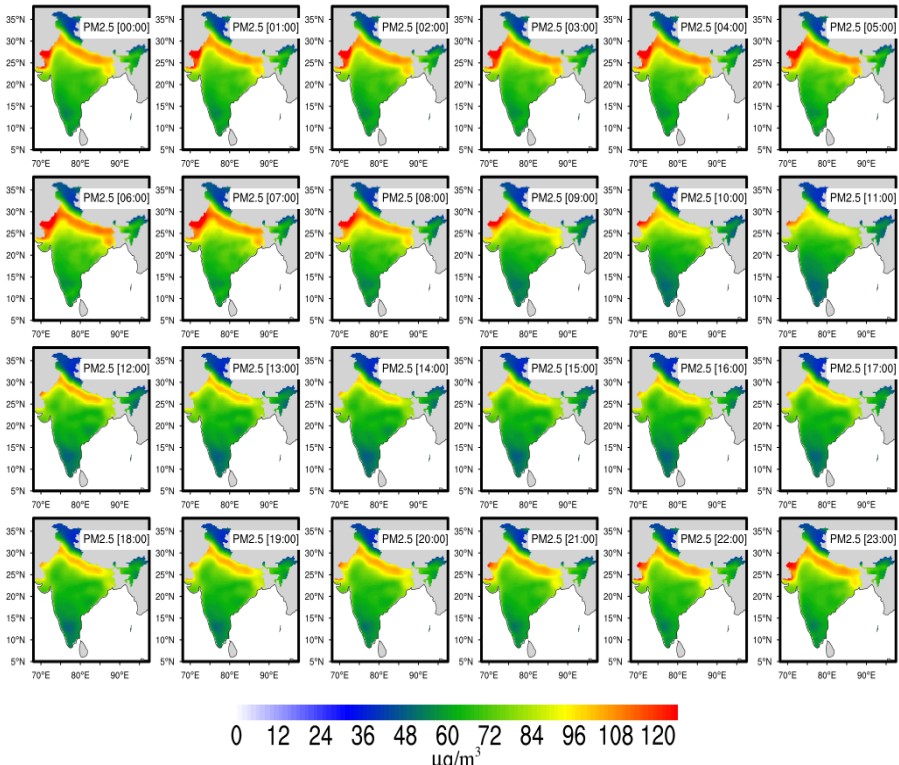

**Figure 5.** Mean annual ambient PM$_{2.5}$ concentration in India for each hour cycle
(00:00 IST represents 00:00-01:00 IST duration).





Manuscript for Atmos. Chem. Phys.

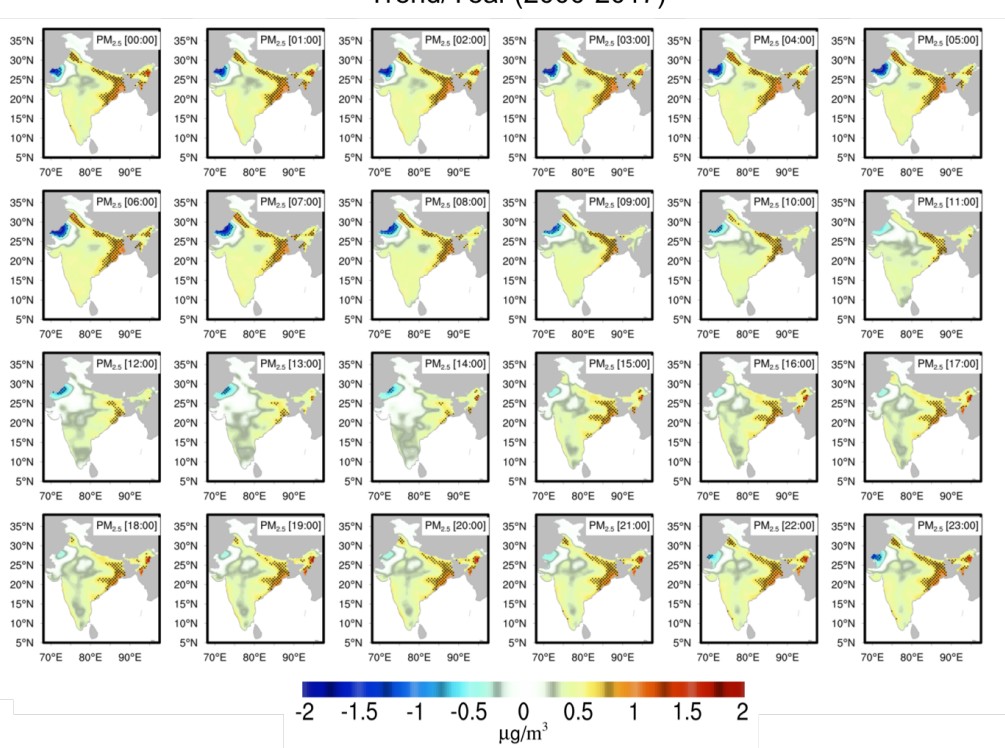



**Figure 6.** Annual trends of ambient PM$_{2.5}$ concentrations (with the hatched regions
showing 95% CI) in India for each hour cycle, during the day (00:00 IST represents
00:00-01:00 IST duration).




Manuscript for Atmos. Chem. Phys.

Trend/Year (2000-2017)

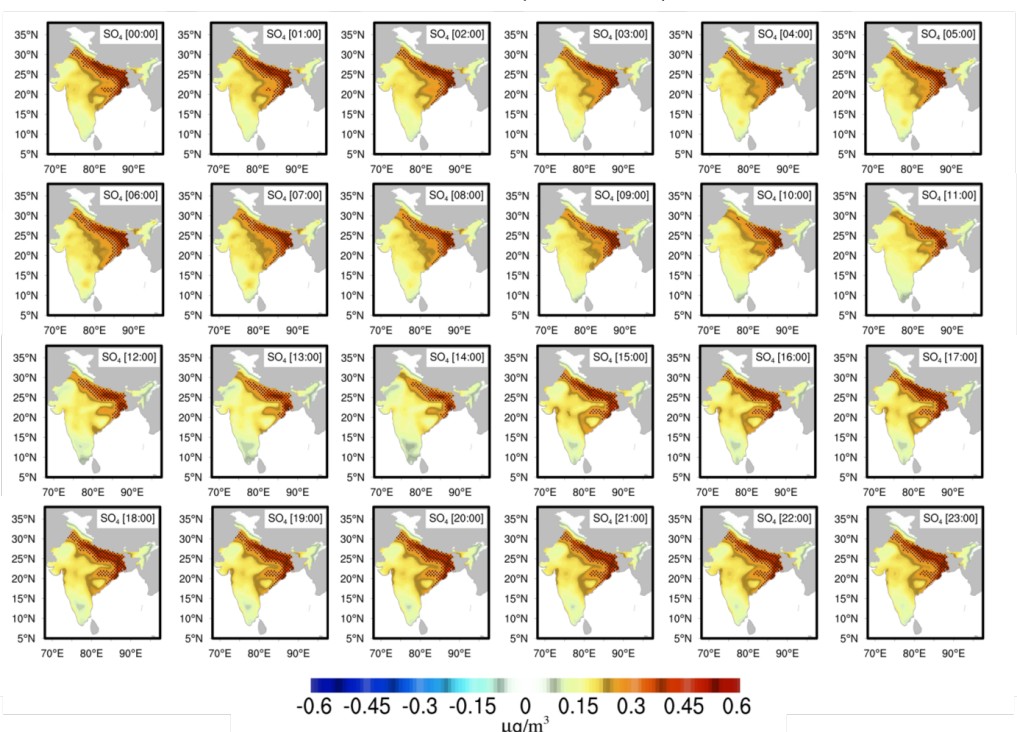

**Figure 7.** Annual trends of ambient sulfate concentrations (with the hatched regions
showing 95% CI) in India for each hour cycle during the day (00:00 IST represents
00:00-01:00 IST duration).
PBL height does not show any significant trend in the monsoon season, which is

reflected in negligible trend in $PM_{2.5}$ concentration (see SI). In the western arid
regions during MAM and in the high altitude regions throughout the year, PBL height
shows an increasing trend. This increasing trend of PBL height, which facilitates
dispersion of aerosols, is perhaps driving the decreasing trend of ambient $PM_{2.5}$
concentration (by $>1.5$ $\mu g$ m$^{-3}$ per year) over the western arid region. This is





Manuscript for Atmos. Chem. Phys.

consistent with the reported decline in dust transport in this region (Pandey et al.,
2017). On the contrary, large PBL height in the high altitude regions would allow the
pollution to be lifted up from the valley beneath more efficiently. This could explain
the increasing trend of $PM_{2.5}$ concentration in these regions during the dry season.

**4. Discussion**
Previously, ambient $PM_{2.5}$ concentrations in India have been assessed either using
data from limited ground-based monitors (Tiwari et al., 2013) or from satellites (van
Donkelaar et al., 2010, 2016; Brauer et al., 2015; Dey et al., 2012; Chowdhury and
Dey, 2016). Neither provides complete spatio-temporal coverage. Even the
geostationary satellite-based AOD product (e.g. Mishra, 2018) is not sufficient to
provide 24-hour coverage. In this work, we propose using MERRA-2 aerosol
reanalysis data to resolve this issue. We document the hour-by-hour changes in $PM_{2.5}$
concentration over 18-year (2000-2017) period for the whole country.
Our results reveal large diurnal amplitude in $PM_{2.5}$ concentration in certain regions
of India and identify the times of maximum and minimum concentration and its
seasonal shift. This explains the underestimation in satellite-based $PM_{2.5}$ estimates
(related to ground-based measurements that cover 24-hour duration) that uses AOD
data from sensors onboard polar orbiting satellites crossing India in the late morning
to early afternoon hours. The regions where the diurnal amplitude is small, satellite-
based estimates of exposures are more representative. We also show that the
increasing trend at annual scale is strongly controlled by increase in $PM_{2.5}$ during
October-February period. This suggests that if the emission during these months can
be controlled, the increasing trend at annual scale can be arrested.
Large increasing trend of $PM_{2.5}$ in the Himalayan foothills is a matter of concern as



transport of pollution to the Himalayan region can adversely affect the cryosphere
(Bali et al., 2017). Another key result is the declining trend in PBL height over a large
part of India especially during the dry season that might have played a major role in
the observed increasing trend in $PM_{2.5}$ concentration. Shallow PBL leads to stagnation
that entraps the pollutants closer to the surface increasing $PM_{2.5}$ concentration. Under
global warming, stagnation events are projected to increase in future over India
(Horton et al., 2014). Therefore, cutting down emission seems to be the only
sustainable solution in addressing air pollution in India.
Although we map hour-by-hour changes in $PM_{2.5}$ concentration in this work, we
cannot identify the major sources at this resolution. Further analyses (of activity and
other secondary data) are required to attribute hourly variation in concentration to any
particular source in any particular region. This study, however, does provide the
opportunity to identify the major sources that can be attributed to the maximum $PM_{2.5}$
(corresponding to the observed peak timing) concentration at a local/regional scale. In
future, integration of these data (spatially and temporally continuous) with ground-
based    measurements    (temporally    continuous)    and    satellite-based    estimates
(temporally discontinuous but can provide information at high spatial resolution) in a
machine-learning framework would perhaps provide the ideal scenario. Nonetheless,
we hope these results will help formulate better air pollution mitigation plans, so that
the national burden of disease attributed to ambient air pollution could be decreased
by evidence-based policy actions at the regional and national levels (Correia et al.,

2013).


*Data availability*. Hourly MERRA-2 $PM_{2.5}$ data can be available on request to the
corresponding author.

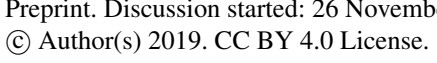



Manuscript for Atmos. Chem. Phys.


*Author contributions*. KB carried out the data processing and analysis. KB and SD
contributed to the writing. SD, DG and KRS contributed to reviewing the article.

*Competing interests*. The authors declare that they have no conflict of interest.

*Acknowledgements*. KB acknowledges financial support from Ministry of
Environment, Forest and Climate Change, Govt. of India through a research grant
(No. 14/10/2014-CC) under the NCAP-COALESCE project. SD acknowledges
financial support from Central Pollution Control Board through a research grant No.
B-12015/101/2019-AS/205. MERRA-2 reanalysis datasets are provided by NASA.
Authors acknowledge Neetu Singh for helping in downloading all CPCB data. SD
acknowledges the DST-FIST grant (SR/FST/ESII-016/2014) for the computing
support.

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
