# Peer review of "Manuscript for Atmos. Chem. Phys. Space-time variability of ambient PM$_{2.5}$ diurnal pattern over India from 18-years (2000-2017) of MERRA-2 reanalysis data"

_Atmospheric Chemistry and Physics, 2019_

## Referee Comment (RC1) · Anonymous Referee #1 · 14 Dec 2019

Air pollution, especially PM2.5, in India is a hot topic. However, studies are hampered by limited availability of data in India. As correctly pointed out in the Introduction that satellite or ground-based measurements have limitations in temporal continuous or spatial coverage. This paper analyses long-term reanalysis MERRA-2 datasets of PM2.5, tries to overcome the limitations and perform analysis of the space-time variability of surface PM2.5 over India during 2000-2017. However, as pointed out in the pre-review by the two referees, the datasets (and quality control) used in this study are not detailed introduced. In this ACPD version, I see little improvements of the description/discussion of the datasets, quality control and validations of the used model results, and representativeness of the observations. I feel less confident about the

drawed conclusion, if the authors do not understand that how are the datasets produced, how is the quality controlled and what are the limitations and representativeness of the data used. I would recommend this manuscript for publication in ACP only if the following concerns can be nicely addressed.

1) Satellite observed AOD550 with MISR, MODIS and AVHRR, plus ground-based AOD data from AERONET were assimilated in the MERRA2 reanalysis dataset. However, all these datasets are column parameters. I understand the assimilation of these datasets can improve the radiative forcing simulation directly. But, how does the assimilation of column parameters improve the simulation of surface PM2.5 concentration in MERRA2? This is not discussed in the paper. More discussion (preferable with some quantifying values) about this would provide more confidence of the model data used, especially for India where very limited surface observations are available or assimilated.

2) Following the above comment, I think the impact of AOD assimilation on surface PM2.5 concentration would be strongly depended on PBL simulation, which is also a key topic analysed in this study. However, how is the PBL simulated and how is the top of PBL defined in the MERRA2; what kinds of meteorological datasets are assimilated in the model to improve the PBL simulation over India; and how good is the performance of the PBL in MERRA2 compared with observations or improved by assimilation, . . .etc? All kinds of these questions are not discussed in the paper.

3) I agree with the second referee that most of the CPCB monitoring are in the urban area: the representativeness of CPCB dataset needs to be carefully discussed. Due to CPCB observations possibly represent the urban condition, how suitable they are for direct comparisons with the MERRA2 global model results with relatively coarse resolution? Based on this thinking, I am kind of agree with the second referee that "the bias correction/calibration methodology is overfitting the model data". This question was not discussed in the ACPD version, I feel we need to think about it more carefully in the next revised version.

[Figure]

4) Furthermore, based on my limited experience with CPCB dataset, it seems the quality of PM2.5 concentration from CPCB is questionable. How is the CPCB dataset quality controlled, this question was raised by the second referee, however, still not addressed. There are some Indian cities have PM2.5 observations from US diplomatic missions, which are generally believed to be of high quality and could help with quality validation of CPCB observations over these cities.

5) MERRA2 dataset is simulated with EDGARv4.2 global emission inventory, as described by the paper. I suppose the EDGAR inventory for year 2012 (the latest one) was used. How well the 2012 inventory represent the condition of the period 2000-2017? As reported by lots of studies, between 2012 and 2017 the Indian emissions have changed a lot. And as described by the MERRA2 aerosol dataset developer (Buchard et al., 2017) that assimilation cannot correct for deficiency due to missing emissions. The uncertainty of emission inventory would propagate to the MERRA2 reanalysis data. And biomass/agriculture burning is believed to be a large contributor of surface PM2.5 over India. How is this burning source considered in the MERRA2 simulation?

6) As described in the paper that OC are secondary aerosols in MERRA2 dataset. I would like to know how is the secondary organic aerosols simulated or represented in the GEOS5/MERRA2 model. Since, OC contributed about half of fine particles mass in Delhi (possibly other IGB regions as well) based on recent observations (Gani et al., 2019). The correct simulation of OC secondary formation processes would be critical for the accuracy of MERRA2 aerosol dataset. Some comments on the validation of OC simulation within MERRA2 would be helpful.

7) Some technical correction: a) line 128. 'three different sensors' should be four sensors in total if count AERONET monitoring as well. b) line 108. I feel the word 'propose' might be inappropriate. This assimilation/reanalysis approach has been widely used over other regions, the contribution of this study is used a reanalysis dataset to analyse the spatial-temporal variation of surface PM2.5 over India.

References: Buchard, V., Randles, C. A., Silva, A. M. d., Darmenov, A., Colarco, P. R., Govindaraju, R., Ferrare, R., Hair, J., Beyersdorf, A. J., Ziemba, L. D., and Yu, H.: The MERRA-2 Aerosol Reanalysis, 1980 Onward. Part II: Evaluation and Case Studies, Journal of Climate, 30, 6851-6872, 10.1175/jcli-d-16-0613.1, 2017. Gani, S., Bhandari, S., Seraj, S., Wang, D. S., Patel, K., Soni, P., Arub, Z., Habib, G., Hildebrandt Ruiz, L., and Apte, J. S.: Submicron aerosol composition in the world's most polluted megacity: the Delhi Aerosol Supersite study, Atmos. Chem. Phys., 19, 6843-6859, 10.5194/acp-19-6843-2019, 2019.
* * *

---

## Referee Comment (RC2) · Anonymous Referee #2 · 20 Dec 2019

Air quality is an important environmental concern. It is more important for countries like India where ambient PM levels are above air quality standard limits. There is a need to understand the temporal and spatial pattern and the sources of pollution over India to take necessary measures.

Studies over India lacks long term analysis of PM2.5 at regional scale. This study claims to present the first space-time variability of ambient PM2.5 diurnal pattern in India for an 18-year (2000-2017) using the bias corrected MERRA2 data. While the objectives of the paper are interesting, the results presented in the paper can be highly uncertain.

[Figure]

Extensive description and evaluation of the MERRA2 Aerosol reanalysis products (1980 onwards) have been presented by Randles et al., (2017) and Buchard et al (2017). In addition to this, Buchard et al (2017) also presented some case studies. Both studies point out and conclude that caveats that must be considered when using this new reanalysis product for future studies of aerosols and their interactions with weather and climate. I am sure this applies Air quality studies as well.

After reading the present manuscript, it gives me impression that authors have not fully understood how the MERRA2 aerosol products has been created, what are the limitations and whether it can be used to address the objectives of the paper. This assessment is in line with the assessment made by referee #1. Following concerns can be addressed before it is accepted for publication

1. Emission annual trend

Firstly, emissions are an important factor to study the spatial and temporal variability and trend. It is important to understand in detail the spatial and temporal scale of the inventories used in the simulation. Authors, in the paper as well as pre-review response, has mentioned that the MERRA-2 products use anthropogenic (EDGARv4.2) and biogenic sulfate (AeroCom Phase II) and carbonaceous aerosols (scaled RETROv2). This gives the impression that most of the anthropogenic emissions are from EDGARv4.2 which is normally available until 2012. However, only anthropogenic SO2 is used from EDGARv4.2 and that is from 1980-2008. The exhaustive list is given in Table 1 of Randles et al., (2017) and also discussed in Randles et al., (2016) where most of the anthropogenic emission is from AeroCom Phase II from 1980-2006. Moreover, Buchard et al (2017) has also mentioned that MERRA-2 anthropogenic emissions vary on a yearly basis, and emissions databases do not extend to 2013 (e.g., 2006 and 2008 are terminal years for anthropogenic OC/BC and SO2 databases, respectively) and same emission is repeated after 2006/2008 until recently Randles et al., (2016).

Therefore, when the terminal years for the anthropogenic emissions are 2006/2008

and constant afterward, can it be used for trend analysis? I am sure one must be very cautious using this data to derive the trend up to 2017.

2. Emissions grid resolution

The native resolution of most of the emissions used for the MERRA2 simulation is over 1deg x 1deg resolution (other than biomass). These emissions datasets are re-gridded to the native model grid. How it could impact the analysis of the paper can be discussed.

3. Hourly analysis

The authors presented the analysis on the hourly scale. The validity of the simulated hourly scale surface PM2.5 concentration lies in the fact how well the meteorology is simulated and the diurnal/hourly profiles used to process the emission. As far as I understand, the seasonal cycle is used (Figure 2.2 of Randles et al., 2016) to speciate annual emission to monthly emissions. There is no mention of diurnal cycle, therefore I assume that no diurnal profile is used for MERRA2 simulations. Moreover, the analysis and comparison of surface PM2.5 across US with MERRA aerosol products presented by Buchard et al (2016 and 2017) were not presented at hourly scale. When the MERRA2 data has been used for hourly scale, then columnar products are used rather than surface products (section 4d of Buchard et al 2017). In a comparison of MERRA2 PM2.5 with surface PM2.5 over North China by Song et al., (2018) has shown that MERRA-2 cannot follow the diurnal variation of PM2.5 but reproduce a good daytime variation of AOD. In this case, once should be concerned about the validity of the results presented at hourly scale.

4. Calculation of PM2.5 from MERRA2.

Authors calculated the PM2.5 by adding up dust and sea-salt in size bins smaller than 2.5m, hydrophilic and hydrophobic OC, BC and sulfate (assuming the entire load is within PM2.5). However a different formula is used in Buchard et al (2016) and Song et

al., (2018). The mass of sulfate is multiplied by 1.375 and OC is multiplied by a factor between 1.2 and 2.6. Authors can comment on why they used unit factor of sulfate and OC in their calculation. Moreover, the SOA ,which dominates in IGP and missing Nitrate aerosols can also be discussed. Also, a larger overestimation of dust and sea salt in MERRA2 (Buchard et al ) can be discussed.

5. CPCB PM2.5 data.

Authors have now provided the list of monitoring stations (Table 1 of suppl material). However the manuscript lacks the description of the CPCB PM2.5 data used in this study. Authors need to provide more information about the methodology/technique/instrument used for the measurement of ambient PM2.5. They need to provide the environment type of each station in table 1, whether they are urban, rural, traffic or background sites. As authors have mentioned that the PM2.5 monitoring started in India by the Central Pollution Control Board (CPCB) in 2008-2009, so all the stations will not have continuous measurements from 2009-2017. Therefore, authors also need to provide the period of valid measurement available and missing period if any. If some stations have continuous measurements from 2009-2017, then there is a chance that the monitoring instrument might have changed. They can mention whether the instrument/technique has changed and how the data has been inter-calibrated. As far as I am aware, CPCB provides the data as measured from the instrument without any quality control. One must do quality control before using the data. Authors may also provide the steps of quality control.

6. Use of CBCP data for this study

To the best of my knowledge and the locations provided by the authors in the table 2. It can be confirmed that most of the stations (it appears that more than 90%) of the stations are in Urban areas. Buchard et al (2016 and 2017) have restricted the analysis of PM2.5 across US over suburban and rural sites because PM2.5 concentrations are generally higher and less uniform in urban areas, such stations are not representative

of the grid-box mean values that MERRA estimates. In this case, I doubt that CPCB urban data is suitable for comparison and bias correction.

7. MERRA2 PM2.5 evaluation and bias estimation

The detailed evaluation of MERRA2 PM2.5 has not been presented in the paper other than mean diurnal plot. Before going for bias correction, it is important to know the temporal and spatial biases in the model. A detailed statistical evaluation has to be presented. The evaluation can be presented for a limited period when most of the data is available. Please refer Song et al., (2018) https://doi.org/10.1016/j.atmosenv.2018.08.012

8. Bias correction methodology.

Although MERRA2 aerosol reanalysis products are better than non-assimilated products, it can have biases, therefore it was calibrated across India (spatially) and during 2009-2017 (temporally) using the CPCB data measured at 80 sites mentioned in supplementary material table 1. To obtain the collocated CBCP and MERRA2 PM2.5, authors have either averaged all CPCB sites within a MERRA-2 grid ($0.5° \times 0.625°$) OR re-grid the MERRA-2 data from 0.5 x 0.625 degree resolution to 0.05 x 0.05 degree resolution and then extracted the PM2.5 values at CPCB coordinates (as per reply to the pre review comments). Authors use 50% CPCB data for bias correction and 50% for validation. Please clarify how the 50% data was selected, was it random or continuous.

First, authors need to address the issues related to CPCB data quality, its availability during 2009-2017 and its spatial representativeness as most of them are in Urban area and are within the same grid. Second, the bias correction methodology needs further clarification as it seems as per the manuscript that authors do two types of bias correction (or calibration). One for in-situ 80 sites and another for the Indian grid. For 80 sites, authors obtain a linear relation between MERRA2 and CPCB PM2.5 and then get the calibration factor as a function of CPCB PM2.5 which is then added to MERRA2

to correct it. (Line 164-171). For Indian grids, authors calculate the calibration factor as a function of MERRA2 2.5 value. To find out the linear regression, authors have binned the data in 500 bins (0-500 ug/m3) (in this way the data becomes independent of the time and location).

For the linear relation used for 80 sites, authors get a liner line with a slope of 0.228 between CPCB and MERRA2. This shows that there is a huge underestimation of MERRA2 PM2.5 most probably because of the use of Urban PM2.5. It is even more surprising liner line between bias (CPCB-MERRA2) has a slope of 0.772. It can be interpreted that model bias has a better correlation then the model estimate. And if the model bias is more than the model estimate then one must rethink before using this data for further analysis.

Finally, the authors find a bias-corrected relation BCM=0.99*CPCB+0.005. Rounding off and further simplification, this equation reduced to BCM=CPCB. It means, all the MERRA2 values are replaced by CPCB values. In this way, authors will certainly get good correlation (0.94) between bias-corrected MERRA2 and validation CPCB PM2.5. Authors can check and report the correlation between validation and the data used for bias correction. By using this methodology, you are overfitting the MEERA2 data. I don't think this is the right way to do the bias correction. There are several papers on bias correction methodology that authors can refer to.

9. Overall comment I have no further comments on the rest of the analysis as it depends on how good is the bias-corrected MERRA2 data. As the moments it appears that 1. MERRA2 PM2.5 may not be suitable for hourly analysis. 2. MARRA2 PM2.5 can not be used for trend analysis because of constant emissions after 2008. 3. MERRA2 aerosols are not suitable for Urban PM2.5 analysis. Because of this, The MERRA2 model bias is more than the MERRA2 estimates. This suggests urban PM2.5 should not be used for bias correction. 4. A robust method of bias correction is required.

Hence, I am less confident that the results presented in the paper are valid. This is

important because the authors claim that these results will help formulate better air pollution mitigation plans by evidence-based policy actions at the regional and national levels. I feel that authors need to be extra cautious and discuss the limitations before publishing these kinds of results. At the moment, it would be appropriate the perform a detailed evaluation of the MERRA2 products over India and present the biases and uncertainties across different temporal scales and geographical regions of India.

10. Some of the minor suggestions

Use either bias corrected MERRA (BCM) or calibrated uniformly. This paper has not been referred: Central Pollution Control Board (CPCB) Ambient air quality statistics for Indian metro cities, Central Pollution Control Board, Zonal Office, Bangalore, 2003. Authors can discuss India specific assimilation used for MERRA2 in detail as indicated by referee#1.

---

## Author Comment (AC1) · 18 Mar 2020

**Referee #1**

*Air pollution, especially PM2.5, in India is a hot topic. However, studies are hampered by limited availability of data in India. As correctly pointed out in the Introduction that satellite or ground-based measurements have limitations in temporal continuous or spatial coverage. This paper analyses long-term reanalysis MERRA-2 datasets of PM2.5, tries to overcome the limitations and perform analysis of the space-time variability of surface PM2.5 over India during 2000-2017. However, as pointed out in the prereview by the two referees, the datasets (and quality control) used in this study are not detailed introduced. In this ACPD version, I see little improvements of the description/discussion of the datasets, quality control and validations of the used model results, and representativeness of the observations. I feel less confident about the drawed conclusion, if the authors do not understand that how are the datasets produced, how is the quality controlled and what are the limitations and representative-ness of the data used. I would recommend this manuscript for publication in ACP only if the following concerns can be nicely addressed.*

We are grateful to the reviewer for providing the insightful comments on our manuscript. We have addressed all the comments and suggestions provided by the reviewer. We discussed the bias correction methods and the MERRA-2 datasets in great details with relevant references. Potential sources of uncertainty and probable limitations are also discussed clearly. Our point-by-point responses for the all the comments are mentioned below.

Comment Response = Red colour
Information in revised MS = blue colour

Comment:1
*Satellite observed AOD550 with MISR, MODIS and AVHRR, plus ground-based AOD data from AERONET were assimilated in the MERRA2 reanalysis dataset. However, all these datasets are column parameters. I understand the assimilation of these datasets can improve the radiative forcing simulation directly. But, how does the assimilation of column parameters improve the simulation of surface PM2.5 concentration in MERRA2?*

Response: We would like to thank the reviewer for raising this issue. We have introduced the following information in the revised MS section 2.1 [Line 141-162] as given below:

   Both the aerosol and meteorology observations jointly assimilated in the GEOS-5 model. The AOD assimilation process includes the cloud screening and bias correction of space-based instruments (e.g. MODIS and AVHRR) with ground-based instrument AERONET. The GEOS-5 model also assimilates the non-biased corrected AOD from MISR (over the bright surface). GEOS-5 assimilates the real-time (for every 3 hours) quality controlled AOD observation by using the local displacement ensemble (LDE) methodology (Buchard et al., 2015, 2017). LDE methodology is introduced to correct the misplaced aerosol plumes by considering the various aerosol properties such as speciation and vertically distribution of aerosols. On the other hand, the regions where LDE is not applied, AOD increment factor has been implemented, which represents a vertical scaling factor, more detail can be found in Buchard et al. 2017.

In addition to this, Buchard et al. (2017) also performed some experiments with assimilation and without assimilation of AOD product over the global region. The results showed that aerosols with the assimilation system provides better estimates of surface $PM_{2.5}$ as they are well correlated with satellite-derived $PM_{2.5}$ from van Donkelaar et al. (2010). However, some discrepancies can be observed due to the lack of the nitrate in the GOCART module and low emission of OC (organic carbon) in MERRA-2 dataset. The carbonaceous MERRA-2 $PM_{2.5}$ are mainly from biomass burning, anthropogenic sources and plant matter (Buchard et al., 2017; Randles et al., 2017). Here, we assume that the contribution of OC concentration from biogenic sources is very less because the aerosol emissions are mainly from biomass burning and anthropogenic sources in Northern India (Gani et al., 2019).

We understand that there could be some uncertainty during the simulation process of $PM_{2.5}$ in MERRA-2 which is also true for any other exposure model, but since this provides hourly dataset, we believe that they can be utilized after applying bias correction. We have included detailed discussion of the dataset and the bias correction technique in the revised MS.

References:
Buchard, V., Da Silva, A.M., Colarco, P.R., Darmenov, A., Randles, C.A., Govindaraju, R., Torres, O., Campbell, J. and Spurr, R., 2015. Using the OMI aerosol index and absorption aerosol optical depth to evaluate the NASA MERRA Aerosol Reanalysis. Atmospheric Chemistry and Physics, 15(10), p.5743.

Buchard, V., da Silva, A.M., Randles, C.A., Colarco, P., Ferrare, R., Hair, J., Hostetler, C., Tackett, J. and Winker, D., 2016. Evaluation of the surface PM2. 5 in Version 1 of the NASA MERRA Aerosol Reanalysis over the United States. Atmospheric environment, 125, pp.100-111.

van Donkelaar, A., Martin, R.V., Brauer, M., Kahn, R., Levy, R., Verduzco, C. and Villeneuve, P.J., 2010. Global estimates of ambient fine particulate matter concentrations from satellite-based aerosol optical depth: development and application. Environmental health perspectives, 118(6), pp.847-855.

Gani, S., Bhandari, S., Seraj, S., Wang, D.S., Patel, K., Soni, P., Arub, Z., Habib, G., Hildebrandt Ruiz, L. and Apte, J.S., 2019. Submicron aerosol composition in the world's most polluted megacity: the Delhi Aerosol Supersite study. Atmospheric Chemistry and Physics, 19(10), pp.6843-6859.

**Comment**:2
*Following the above comment, I think the impact of AOD assimilation on surfacePM2.5 concentration would be strongly depended on PBL simulation, which is also a key topic analysed in this study. However, how is the PBL simulated and how is the top of PBL defined in the MERRA2; what kinds of meteorological datasets are assimilated in the model to improve the PBL simulation over India; and how good is the performance of the PBL in MERRA2 compared with observations or improved by assimilation,...etc? All kinds of these questions are not discussed in the paper.*

**Response**: We would like to thank the reviewer for this suggestion. We have incorporated the information in the revised MS [Line 247-269] as given below:

According to Rienecker et al. (2011), two types of schemes are introduced in the GEOS-5 model for simulating the atmospheric boundary layer in the MERRA-2 reanalysis dataset. The first scheme (Louis et al., 1982) is based on the planetary stable condition in which no planetary boundary layer of clouds involved. The second scheme (Lock et al., 2000) is based on the unstable or cloud-topped planetary boundary layer condition involved. Additionally, the GEOS-5 model uses two more schemes based on orographic conditions such as orographic gravity wave drag (McFarlane 1987) and non-orographic waves (Garcia and Boville 1994).

MERRA PBL heights are also diagnosed by the turbulence parameterization in the atmospheric general circulation model, based on the eddy diffusivity coefficient for heat. The PBL height is diagnosed as the lowest level at which the diffusion coefficients drop below a value of 1 $m^2$/s. There are various meteorological variables (temperature, wind, and humidity) also introduced in the GEOS-5 model to assimilate the PBL heights. Regarding the performance of the GEOS-5 model for PBL height, Jordan et al. (2010) observed a good ($r = 0.7$) correlation of PBL height between CALIOP satellite and GEOS-5 model over the western hemisphere (e.g. American and African region). However, some disagreement was also observed over the equatorial Pacific Ocean, where GEOS-5 PBL height is lower than the CALIOP PBL height. In addition to PBL height, the other physical processes such as aerosol mixing and hygroscopic growth etc. could also contribute to the uncertainty in the aerosol model (Randles et al., 2017, Schutgen et al. 2010). With the calibration, we were able to reduce this low bias substantially.

*References:*
*Jordan, N.S., Hoff, R.M. and Bacmeister, J.T., 2010. Validation of Goddard Earth Observing System-version 5 MERRA planetary boundary layer heights using CALIPSO. Journal of Geophysical Research: Atmospheres, 115(D24).*

*McFarlane, N.A., 1987. The effect of orographically excited gravity wave drag on the general circulation of the lower stratosphere and troposphere. Journal of the atmospheric sciences, 44(14), pp.1775-1800.*

*Garcia, R.R. and Boville, B.A., 1994. "Downward control" of the mean meridional circulation and temperature distribution of the polar winter stratosphere. Journal of the atmospheric sciences, 51(15), pp.2238-2245.*

*Randles, C. A., Da Silva, A. M., Buchard, V., Darmenov, A., Colarco, P. R., Aquila, V., Bian, H., Nowottnick, E. P., Pan, X., Smirnov, A., Yu, H., and Govindaraju, R.: The MERRA-2 Aerosol Assimilation. Technical Report Series on Global Modeling and Data Assimilation 44, NASA Global Modeling and Assimilation Office. [Available online at https://gmao.gsfc.nasa.gov/reanalysis/MERRA-2/docs/.], 2016.*

*Schutgens, N.A.J., Miyoshi, T., Takemura, T. and Nakajima, T., 2010. Sensitivity tests for an ensemble Kalman filter for aerosol assimilation. Atmospheric Chemistry & Physics Discussions, 10(3).*

Comment:3
*I agree with the second referee that most of the CPCB monitoring are in the urban area: the representativeness of CPCB dataset needs to be carefully discussed. Due to CPCB observations possibly represent the urban condition, how suitable they are for direct comparisons with the MERRA2 global model results with relatively coarse resolution? Based on this thinking, I am kind of agree with the second referee that "the bias correction/calibration methodology is overfitting the model data". This question was not discussed in the ACPD version, I feel we need to think about it more carefully in the next revised version*

Response: We appreciate the reviewer for pointing out this issue. We have calculated the MERRA-2 PM$_{2.5}$ by using some scaling factor (Line 173) and further modified the previous bias correction method in the revised MS. This has addressed the overfitting issue pointed out by both reviewers. We personally went through the CPCB data for quality check. The detailed information of the bias correction can be found in the revised MS [Line 198-222] that is given as:

**2.3 Calibration of MERRA-2 PM$_{2.5}$ with CPCB**

We calibrated hourly MERRA-2 PM$_{2.5}$ with coincident PM$_{2.5}$ data from 75 CPCB sites across the country for the period 2009-2017, as CPCB data are available only for these periods. The uncalibrated MERRA-2 PM$_{2.5}$ shows a correlation of 0.57 (significant at 95% CI) with coincident in-situ PM$_{2.5}$ (left panel of Figure 1). For bias correction, we used the percentile based bias correction methodology. We divided the MERRA-2 data at 10 percent interval and then calculated the relationship ($r = 0.9$) between median bias at every 10 percentile ranges between the two datasets (central panel of Figure 1). Then we adjusted MERRA-2 data with the calibration factors of the respective percentile ranges. Bias-corrected MERRA-2 at every grid (Figure S1) and median PM$_{2.5}$ at every 50 $\mu$g m$^{-3}$ interval (right panel of Figure 1) show improved correlation with the in-situ data. We note that MERRA-2 PM$_{2.5}$ is still underestimated at very high concentration (i.e. >300 $\mu$g m$^{-3}$); but since most of the country does not have any ground-based monitoring, we proceeded with our analysis with the calibrated MERRA-2 PM$_{2.5}$ to examine the diurnal pattern over India.

[Figure]

**Figure 1.** Scatter plot between (left) uncalibrated MERRA-2 and in-situ PM$_{2.5}$ data, (middle) median bias in MERRA-2 and in-situ PM$_{2.5}$ at every 10 percentile ranges, and (right) calibrated

median PM$_{2.5}$ from MERRA-2 and in-situ at every 50 $\mu$g m$^{-3}$ interval. Spatial and temporal matching is done by averaging data from all ground-based monitoring sites falling within a single MERRA-2 grid for 1-hr duration.

Additionally, we would like to report that, modeling oriented studies either underestimate or overestimate the values with respect to the ground-based or space-based observational values. However, it is also true that models are also capable of capturing the more or less same spatial/temporal pattern with the observational data (in-situ and remote sensing). Therefore, in our study, we would like to point out that the MERRA-2 dataset has also shown the more or less same PM2.5 diurnal with in-situ CPCB over the Indian region.

Comment:4
*Furthermore, based on my limited experience with CPCB dataset, it seems the quality of PM$_{2.5}$ concentration from CPCB is questionable. How is the CPCB dataset quality controlled, this question was raised by the second referee, however, still not addressed. There are some Indian cities have PM$_{2.5}$ observations from US diplomatic missions, which are generally believed to be of high quality and could help with quality validation of CPCB observations over these cities*

Response:
It is difficult to objectify accuracy of the ground-based sensors based on perception. There is no study documenting whether the embassy monitors are more accurate than CPCB. Moreover, embassy monitoring stations (assumed to have high quality PM$_{2.5}$ data) operate the instrument at the embassy ground only which are located in urban centers (Pant et al., 2018). Therefore, they don't provide heterogenous environment. The other monitoring networks such as SAFAR and MAPAN are not available in the public domain.

We took caution in handling CPCB data. We personally went through the entire raw data, and eliminate data from the period where it showed spurious and wild fluctuations. We understand the importance of data quality by the regulatory agencies, but we cannot do more than this. WHO and exposure data used in GBD (Shaddick et al., 2018) also had to calibrate with CPCB network.

*Reference:*
*Pant, P., Lal, R.M., Guttikunda, S.K., Russell, A.G., Nagpure, A.S., Ramaswami, A. and Peltier, R.E., 2019. Monitoring particulate matter in India: recent trends and future outlook. Air Quality, Atmosphere & Health, 12(1), pp.45-58.*

Comment:5
*MERRA2 dataset is simulated with EDGARv4.2 global emission inventory, as described by the paper. I suppose the EDGAR inventory for year 2012 (the latest one) was used. How well the 2012 inventory represent the condition of the period 2000-2017? As reported by lots of studies, between 2012 and 2017 the Indian emissions have changed a lot. And as described by the MERRA2 aerosol dataset developer (Buchard et al., 2017) that assimilation cannot correct for deficiency due to missing emissions. The uncertainty of emission inventory would propagate to the MERRA2 reanalysis data. And biomass/agriculture burning is believed to be a large contributor of surface PM2.5 over India. How is this burning source considered in the MERRA2 simulation.*

Response:

MERRA-2 dataset includes different emission inventories (Table 1), in-situ and satellite datasets. The years in which emission inventories (EDGAR) are not available, different satellites (MODIS-Terra and Aqua, MISR) and in-situ data (AERONET) are utilized to simulate the aerosols in the MERRA-2 datasets along with fire emission inventories (QFED and GFED). We agree that there are obvious uncertainty in the simulations as emission is not periodically updated, but this is true for any modelling exercise. Globally, no emission inventory is updated annually. We believe that since MERRA-2 assimilates satellite and ground-based data, it should capture the temporal heterogeneity/variation. Further, our bias correction has brought the data closer to the observations.

Table 1. According to Randles et al. (2017), the aerosol species and their sources and inventories used are given as

| Aerosol Type | Sources |
| --- | --- |
| Dust | Wind driven |
| Sea Salt | Wind Driven |
| Volcanic SO2 | AeroCom Phase II |
| Biomass Burning | Scaled RETROv2 |
| SO2, SO4, POM and BC | Scaled GFEDv3.1 QFED 2.4-r6 |
| Anthropogenic SO2 | EDGARv4.2 (Energy + non Energy) |
| Anthropogenic SO4, BC, and POM | AeroCom Phase II |

We also thank the reviewer for asking about the burning source in the MERRA2 simulation. We have incorporated the information in the revised MS [Line 129-138] is given as:

Randles et al., (2017) and Buchard et al. (2017) also mentioned that Quick Fire Emissions Dataset (QFED) emission inventory (based on the MODIS fire radiative power) has been used from 2010 onward, but from the period of 1997-2009, the Global Fire Emission Dataset (GFEDv3.1) emission inventory has been introduced to simulate the aerosol emission from biomass burning in MERRA-2 reanalysis dataset (Darmenov and da Silva 2015; Randerson et al., 2006; van der Werf et al., 2006). Some scaling factor was also introduced in the QFED to minimize the uncertainty among the biomass burning emission inventories (Buchard et al., 2017). So, use of two different inventories for different time period could also be responsible for the uncertainty in fire-based MERRA-2 $PM_{2.5}$.

*References:*

*Buchard, V., da Silva, A.M., Randles, C.A., Colarco, P., Ferrare, R., Hair, J., Hostetler, C., Tackett, J. and Winker, D., 2016. Evaluation of the surface PM2. 5 in Version 1 of the NASA MERRA Aerosol Reanalysis over the United States. Atmospheric environment, 125, pp.100-111.*

*Darmenov, A. and da Silva, A., 2013. The quick fire emissions dataset (QFED)–documentation of versions 2.1, 2.2 and 2.4. NASA Technical Report Series on Global Modeling and Data Assimilation, NASA TM-2013-104606, 32, p.183.*

*Randles, C.A., Da Silva, A.M., Buchard, V., Colarco, P.R., Darmenov, A., Govindaraju, R., Smirnov, A., Holben, B., Ferrare, R., Hair, J. and Shinozuka, Y., 2017. The MERRA-2 aerosol reanalysis, 1980 onward. Part I: System description and data assimilation evaluation. Journal of Climate, 30(17), pp.6823-6850.*

*Randerson JT, Liu H, Flanner MG, Chambers SD, Jin Y, Hess PG, Pfister G, Mack MC, Treseder KK, Welp LR, Chapin FS. The impact of boreal forest fire on climate warming. science. 2006 Nov 17;314(5802):1130-2.*

*Van Der Werf, G.R., Randerson, J.T., Giglio, L., Collatz, G.J., Kasibhatla, P.S. and Arellano Jr, A.F., 2006. Interannual variability of global biomass burning emissions from 1997 to 2004.*

Commnet:6
*As described in the paper that OC are secondary aerosols in MERRA2 dataset. I would like to know how is the secondary organic aerosols simulated or represented in the GEOS5/MERRA2 model. Since, OC contributed about half of fine particles mass in Delhi (possibly other IGB regions as well) based on recent observations (Gani et al., 2019). The correct simulation of OC secondary formation processes would be critical for the accuracy of MERRA2 aerosol dataset. Some comments on the validation of OC simulation within MERRA2 would be helpful.*

Response:6
We would like to thank the reviewer for pointing out this comment. We have modified the revised MS [Line 158-162] and incorporated as:

In the MERRA-2 dataset, the carbonaceous $PM_{2.5}$ are from biomass burning, anthropogenic sources and plant matter (Buchard et al., 2017; Randles et al., 2017). Here, we are assuming that the contribution of OC concentration from biogenic sources is very less because the aerosol emissions are mainly from biomass burning and anthropogenic sources in Northern India (Gani et al., 2019).

Regarding the validation of OC emissions from MERRA-2, Bali et al., 2017 compared the OC emissions from MERRA-2 with ECMWF-GFAS (Global Fire Assimilation System) datasets. The study observed the same spatial distribution of OC emission in both the reanalysis datasets but with the high estimation of OC values in MERRA-2.

**References:**
*Buchard, V., da Silva, A.M., Randles, C.A., Colarco, P., Ferrare, R., Hair, J., Hostetler, C., Tackett, J. and Winker, D., 2016. Evaluation of the surface PM2. 5 in Version 1 of the NASA MERRA Aerosol Reanalysis over the United States. Atmospheric environment, 125, pp.100-111.*

*Randles, C.A., Da Silva, A.M., Buchard, V., Colarco, P.R., Darmenov, A., Govindaraju, R., Smirnov, A., Holben, B., Ferrare, R., Hair, J. and Shinozuka, Y., 2017. The MERRA-2 aerosol*

*reanalysis, 1980 onward. Part I: System description and data assimilation evaluation. Journal of Climate, 30(17), pp.6823-6850.*

*Gani, S., Bhandari, S., Seraj, S., Wang, D.S., Patel, K., Soni, P., Arub, Z., Habib, G., Hildebrandt Ruiz, L. and Apte, J.S., 2019. Submicron aerosol composition in the world's most polluted megacity: the Delhi Aerosol Supersite study. Atmospheric Chemistry and Physics, 19(10), pp.6843-6859.*

Comment:7

*Some technical correction: a) line 128. 'three different sensors' should be four sensors in total if count AERONET monitoring as well. b) line 108. I feel the word 'propose' might be inappropriate. This assimilation/reanalysis approach has been widely use dover other regions, the contribution of this study is used a reanalysis dataset to analyse the spatial-temporal variation of surface PM2.5 over India.*

Response:7
    a) has changed in the revised MS as per referee suggestion.
    b) has changed in the revised MS as per referee suggestion.

---

## Author Comment (AC2) · 18 Mar 2020

*Air quality is an important environmental concern. It is more important for countries like India where ambient PM levels are above air quality standard limits. There is a need to understand the temporal and spatial pattern and the sources of pollution over India to take necessary measures.*

*Studies over India lacks long term analysis of PM2.5 at regional scale. This study claims to present the first space-time variability of ambient PM2.5 diurnal pattern in India for an 18-year (2000-2017) using the bias corrected MERRA2 data. While the objectives of the paper are interesting, the results presented in the paper can be highly uncertain.*

*Extensive description and evaluation of the MERRA2 Aerosol reanalysis products(1980 onwards) have been presented by Randles et al., (2017) and Buchard et al(2017). In addition to this, Buchard et al (2017) also presented some case studies. Both studies point out and conclude that caveats that must be considered when using this new reanalysis product for future studies of aerosols and their interactions with weather and climate. I am sure this applies Air quality studies as well. After reading the present manuscript, it gives me impression that authors have not fully understood how the MERRA2 aerosol products has been created, what are the limitations and whether it can be used to address the objectives of the paper. This assessment is in line with the assessment made by referee #1. Following concerns can be addressed before it is accepted for publication.*

We are grateful to the reviewer for providing the insightful comments on our manuscript. We have addressed all the comments and suggestions provided by the reviewer. Our point-by-point responses for the all the comments are mentioned below.

Comment Response = Red colour
Information in revised MS = blue colour

*Comment #1 Emission annual trend*

*Firstly, emissions are an important factor to study the spatial and temporal variability and trend. It is important to understand in detail the spatial and temporal scale of the inventories used in the simulation. Authors, in the paper as well as pre-review response, has mentioned that the MERRA-2 products use anthropogenic (EDGARv4.2) and bio-genic sulfate (AeroCom Phase II) and carbonaceous aerosols (scaled RETROv2). This gives the impression that most of the anthropogenic emissions are from EDGARv4.2 which is normally available until 2012. However, only anthropogenic SO2 is used from EDGARv4.2 and that is from 1980-2008. The exhaustive list is given in Table 1 of Randles et al., (2017) and also discussed in Randles et al., (2016) where most of the anthropogenic emission is from AeroCom Phase II from 1980-2006. Moreover, Buchard et al (2017) has also mentioned that MERRA-2 anthropogenic emissions vary on a yearly basis, and emissions databases do not extend to 2013 (e.g., 2006 and 2008 are terminal years for anthropogenic OC/BC and SO2 databases, respectively) and same emission is repeated after 2006/2008 until recently Randles et al., (2016).*

*Therefore, when the terminal years for the anthropogenic emissions are 2006/2008 and constant afterward, can it be used for trend analysis? I am sure one must be very cautious using this data to derive the trend up to 2017.*

Response:
We would like to thank the reviewer for pointing out this issue. We understand that emission input every year is ideal, but no inventory has annual update available for use by the models. On the other hand, MERRA-2 has inputs from 4 different sensors (MODIS-Terra, Aqua, MISR, AVHRR, and AERONET) with the emission inventories. These sensors are highly reliable and accepted for different air quality studies over a global scale. We believe that since aerosol data are assimilated in the models, it addresses the non-availability of emission update indirectly. Our evaluation shows that broad patterns match with the observations, but MERRA-2 underestimates. This low bias has been corrected to a great extent by our bias correction.

*Comment #2 Emissions grid resolution*

*The native resolution of most of the emissions used for the MERRA2 simulation is over 1deg x 1deg resolution (other than biomass). These emissions datasets are re-gridded to the native model grid. How it could impact the analysis of the paper can be discussed.*

Response: We believe that this would not affect our broader conclusions. Van Donkelaar $PM_{2.5}$ database that was extensively used in successive GBD studies was derived from simulations at coarser resolutions interpolated to finer resolutions. We added this line in the discussion part [Line 488-491].

*Comment #3 Hourly analysis*

*The authors presented the analysis on the hourly scale. The validity of the simulated hourly scale surface PM2.5 concentration lies in the fact how well the meteorology is simulated and the diurnal/hourly profiles used to process the emission. As far as I understand, the seasonal cycle is used (Figure 2.2 of Randles et al., 2016) to speciate annual emission to monthly emissions. There is no mention of diurnal cycle, therefore I assume that no diurnal profile is used for MERRA2 simulations. Moreover, the analysis and comparison of surface PM2.5 across US with MERRA aerosol products presented by Buchard et al (2016 and 2017) were not presented at hourly scale. When the MERRA2 data has been used for hourly scale, then columnar products are used rather than surface products (section 4d of Buchard et al 2017). In a comparison of MERRA2 PM2.5 with surface PM2.5 over North China by Song et al., (2018) has shown that MERRA-2 cannot follow the diurnal variation of PM2.5 but reproduce a good daytime variation of AOD. In this case, once should be concerned about the validity of the results presented at hourly scale.*

Response:
We would like to thank the reviewer for providing this suggestion. We have modify our MS as per reviewer suggestion. The inserted information in revised MS section 2.3 [Line 198-243] is given as:

**2.3 Calibration of MERRA-2 $PM_{2.5}$ with CPCB**

We calibrated hourly MERRA-2 $PM_{2.5}$ with coincident $PM_{2.5}$ data from 75 CPCB sites across the country for the period 2009-2017, as CPCB data are available only for these periods.

The uncalibrated MERRA-2 PM$_{2.5}$ shows a correlation of 0.57 (significant at 95% CI) with coincident in-situ PM$_{2.5}$ (left panel of Figure 1). For bias correction, we used the percentile based bias correction methodology. We divided the MERRA-2 data at 10 percent interval and then calculated the relationship ($r = 0.9$) between median bias at every 10 percentile ranges between the two datasets (central panel of Figure 1). Then we adjusted MERRA-2 data with the calibration factors of the respective percentile ranges. Bias-corrected MERRA-2 at every grid (Figure S1) and median PM$_{2.5}$ at every 50 $\mu$g m$^{-3}$ interval (right panel of Figure 1) show improved correlation with the in-situ data. We note that MERRA-2 PM$_{2.5}$ is still underestimated at very high concentration (i.e. >300 $\mu$g m$^{-3}$); but since most of the country does not have any ground-based monitoring, we proceeded with our analysis with the calibrated MERRA-2 PM$_{2.5}$ to examine the diurnal pattern over India.

[Figure]

**Figure 1.** Scatter plot between (left) uncalibrated MERRA-2 and in-situ PM$_{2.5}$ data, (middle) median bias in MERRA-2 and in-situ PM$_{2.5}$ at every 10 percentile ranges, and (right) calibrated median PM$_{2.5}$ from MERRA-2 and in-situ at every 50 $\mu$g m$^{-3}$ interval. Spatial and temporal matching is done by averaging data from all ground-based monitoring sites falling within a single MERRA-2 grid for 1-hr duration.

**2.4 Diurnal patterns in calibrated MERRA-2 and CPCB data**

We have further analysed the diurnal variation of PM$_{2.5}$ in each season (JF, MAM, JJAS and OND) from calibrated MERRA-2 and CPCB data (Figure 2). The black, blue and red lines represent the mean of all 75 stations (2009-2017) from CPCB, uncalibrated MERRA-2, and calibrated MERRA-2, respectively.

[Figure]

**Figure 2**. Diurnal variation of PM$_{2.5}$ of CPCB, uncalibrated MERRA-2 (without bias correction, WBC) and calibrated MERRA-2 (after correcting bias, ABC) for OND, JF, MAM, and JJAS period. The shaded regions represent ±1σ around seasonal mean across India.

Two observations are notable. Firstly, calibrated MERRA-2 PM$_{2.5}$ is much closer to CPCB data relative to the uncalibrated MERRA-2 data across 24-hr period. Secondly, the calibrated MERRA-2 PM$_{2.5}$ is able to mimic the observed diurnal pattern in all the seasons except the winter (JF) when the low bias of calibrated MERRA-2 PM$_{2.5}$ is prominent. Capturing extreme pollution level and very shallow boundary layer is always challenging for the models. MERRA-2 captures the variation of high (i.e. early morning or late evening) and low PM$_{2.5}$ concentration (noontime to 4 p.m.).

*Comment #4 Calculation of PM2.5 from MERRA2.*

*Authors calculated the PM2.5 by adding up dust and sea-salt in size bins smaller than2.5m, hydrophilic and hydrophobic OC, BC and sulfate (assuming the entire load is within PM2.5). However a different formula is used in Buchard et al (2016) and Song et al., (2018). The mass of sulfate is multiplied by 1.375 and OC is multiplied by a factor between 1.2 and 2.6. Authors can comment on why they used unit factor of sulfate and OC in their calculation. Moreover, the SOA ,which dominates in IGP and missing Nitrate aerosols can also be discussed. Also, a larger overestimation of dust and sea-salt in MERRA2 (Buchard et al ) can be discussed.*

Response:
We thank the reviewer for pointing out this issue. We have modified the calculation of PM$_{2.5}$ in the revised MS. The bias correction method is also revised based on this. The following information are introduced in revised MS [Line 172-181] as:

In our study, the MERRA-2 total PM$_{2.5}$ is calculated as

$$\text{MERRA-2 PM}_{2.5} = [\text{Dust}_{2.5}] + [\text{SS}_{2.5}] + [\text{BC}] + 1.6 \times [\text{OC}] + 1.375 \times [\text{SO}_4].$$

To obtain total PM$_{2.5}$, we simply add up dust and sea-salt in size bins smaller than 2.5 $\mu$m, hydrophilic and hydrophobic OC, BC and sulfate (assuming the entire load is within PM$_{2.5}$). Sulfate concentration is present in the form of neutralized ammonium sulfate [(NH$_4$)$_2$SO$_4$] in MERRA-2 datasets, so a factor of 1.375 (Buchard et al., 2016, Song et al., 2018) is used to obtain the 'true' sulfate concentration. The particulate organic matter (POM) is estimated by multiplying OC by a factor 1.6 (Ram et al., 2012), which accounts for contributions from other elements associated with other organic matter.

*Song, Z., Fu, D., Zhang, X., Wu, Y., Xia, X., He, J., Han, X., Zhang, R. and Che, H., 2018. Diurnal and seasonal variability of PM2. 5 and AOD in North China plain: Comparison of MERRA-2 products and ground measurements. Atmospheric Environment, 191, pp.70-78.*

*Buchard, V., da Silva, A.M., Randles, C.A., Colarco, P., Ferrare, R., Hair, J., Hostetler, C., Tackett, J. and Winker, D., 2016. Evaluation of the surface PM2. 5 in Version 1 of the NASA MERRA Aerosol Reanalysis over the United States. Atmospheric environment, 125, pp.100-111.*

*Ram, K., Sarin, M.M. and Tripathi, S.N., 2012. Temporal trends in atmospheric PM2. 5, PM10, elemental carbon, organic carbon, water-soluble organic carbon, and optical properties: impact of biomass burning emissions in the Indo-Gangetic Plain. Environmental science & technology, 46(2), pp.686-695.*

*Comment #5 CPCB PM2.5 data.*
*Authors have now provided the list of monitoring stations (Table 1 of suppl material). However the manuscript lacks the description of the CPCB PM2.5 data used in this study. Authors need to provide more information about the methodology/technique/instrument used for the measurement of ambient PM2.5. They need to provide the environment type of each station in table 1, whether they are urban, rural, traffic or background sites. As authors have mentioned that the PM2.5 monitoring started in India by the Central Pollution Control Board (CPCB) in 2008-2009, so all the stations will not have continuous measurements from 2009-2017. Therefore, authors also need to provide the period of valid measurement available and missing period if any. If some stations have continuous measurements from 2009-2017, then there is a chance that the monitoring instrument might have changed. They can mention whether the instrument/technique has changed and how the data has been inter-calibrated. As far as I am aware, CPCB provides the data as measured from the instrument without any quality control. One must do quality control before using the data. Authors may also provide the steps of quality control.*

Response: It is difficult to objectify accuracy of the ground-based sensors based on perception. There is no study documenting whether the embassy monitors are more accurate than CPCB. Moreover, embassy monitoring stations (assumed to have high quality $PM_{2.5}$ data) operate the instrument at the embassy ground only which are located in urban centers (Pant et al., 2018). Therefore, they don't provide heterogenous environment. The other monitoring networks such as SAFAR and MAPAN are not available in the public domain. We took caution in handling CPCB data. We personally went through the entire raw data, and eliminate data from the period where it showed spurious and wild fluctuations. We understand the importance of data quality by the regulatory agencies, but we cannot do more than this. WHO and exposure data used in GBD (Shaddick et al., 2018) also had to calibrate with CPCB network.

We have added the description regarding CPCB dataset in the revised MS [Line 182-197]. The following lines are introduced as:

**2.2 CPCB (Central Pollution Control Board) dataset**

CPCB is the regulatory organization under the Ministry of Environment, Forest and Climate Change (MoEFCC), Government of India responsible for the environmental management. CPCB launched a national air quality monitoring program (NAMP) to provide pollution data (i.e. $PM_{2.5}$, $PM_{10}$, $SO_2$, $NO_2$) using gravimetric measurement since 2009. CPCB has formulated the $PM_{2.5}$ standards for the Indian region in 2009 - 60 $\mu$g m$^{-3}$ for 24-h and 40

$\mu g\ m^{-3}$ for annual concentration, which are generally higher than WHO guideline of 25 $\mu g\ m^{-3}$ for 24-h and 10 $\mu g\ m^{-3}$ for annual (Pant et al., 2018). CPCB also established a network with Continuous Automatic Air Quality Monitoring Station (CAAQMS) throughout major cities (with population > 1 million) in India. CPCB stations are generally installed in residential and industrial areas are limited to the urban areas (Gordon et al., 2018). The documents regarding sampling procedures, measurement methods, and QA/QC (quality assurance/quality control) procedures are available in CPCB (2003, 2011). We performed additional QA/QC check with the data checking abrupt changes and abnormally high or low values within 1-hour duration and used the data we feel confident about.

References:

*Pant, P., Lal, R.M., Guttikunda, S.K., Russell, A.G., Nagpure, A.S., Ramaswami, A. and Peltier, R.E., 2019. Monitoring particulate matter in India: recent trends and future outlook. Air Quality, Atmosphere & Health, 12(1), pp.45-58.*

*Gordon, T., Balakrishnan, K., Dey, S., Rajagopalan, S., Thornburg, J., Thurston, G., Agrawal, A., Collman, G., Guleria, R., Limaye, S. and Salvi, S., 2018. Air pollution health research priorities for India: Perspectives of the Indo-US Communities of Researchers. Environment international, 119, p.100.*

*CPCB (2003) Guidelines for ambient air quality monitoring; series: NAAQMS/ ... /2003-04. Central Pollution Control Board. https:// tinyurl.com/y8r54a3z. Accessed 12 Nov 2017.*

*CPCB (2011) Guidelines for the measurement of ambient air pollutants volume-II. Central Pollution Control Board. https://tinyurl.com/ ycjs4h8j. Accessed 12 Nov 2017.*

*Comment#6 Use of CBCP data for this study*

*To the best of my knowledge and the locations provided by the authors in the table2. It can be confirmed that most of the stations (it appears that more than 90%) of the stations are in Urban areas. Buchard et al (2016 and 2017) have restricted the analysis of PM2.5 across US over suburban and rural sites because PM2.5 concentrations are generally higher and less uniform in urban areas, such stations are not representative of the grid-box mean values that MERRA estimates. In this case, I doubt that CPCB urban data is suitable for comparison and bias correction.*

Response: Apart from CPCB $PM_{2.5}$ dataset, we don't have any publicly available in-situ $PM_{2.5}$ observational dataset in terms of spatial and temporal resolution across the whole Indian region. So, we had to rely on the available CPCB urban $PM_{2.5}$ dataset for the validation part with the MERRA-2 dataset. We took caution in handling CPCB data. We personally went through the entire raw data, and eliminate data from the period where it showed spurious and wild fluctuations. We understand the importance of data quality by the regulatory agencies, but we cannot do more than this. WHO and exposure data used in GBD (Shaddick et al., 2018) also had to calibrate with CPCB network.

After processing the CPCB data, we observed that MERRA-2 can provide the best possible

result if we apply the percentile (10%) based bias correcting method w.r.t CPCB (Figure 1). Considering the validation results of Section 2.4 in the revised MS (Line 237-243), we can argue that MERRA-2 can be used for urban sites as well, for the Indian region. Two observations are notable. Firstly, calibrated MERRA-2 $PM_{2.5}$ is much closer to CPCB data relative to the uncalibrated MERRA-2 data across 24-hr period. Secondly, the calibrated MERRA-2 $PM_{2.5}$ is able to mimic the observed diurnal pattern in all the seasons except the winter (JF) when the low bias of calibrated MERRA-2 $PM_{2.5}$ is prominent. Capturing extreme pollution level and very shallow boundary layer is always challenging for the models. MERRA-2 captures the variation of high (i.e. early morning or late evening) and low $PM_{2.5}$ concentration (noontime to 4 p.m.). We also observed the same pattern for different states (Supp. R1-R3) which covers the north, west, south, and east part of India. We observed the high and low pollution during winters and pre-monsoon/monsoon respectively for each state respectively.

*Comment#7 MERRA2 PM2.5 evaluation and bias estimation*

*The detailed evaluation of MERRA2 PM2.5 has not been presented in the paper other than mean diurnal plot. Before going for bias correction, it is important to know the temporal and spatial biases in the model. A detailed statistical evaluation has to be presented. The evaluation can be presented for a limited period when most of the data is available. Please refer Song et al., (2018)https://doi.org/10.1016/j.atmosenv.2018.08.012.*

Response: We thank the reviewer for pointing out this suggestion. We have introduced some validation part of mean (75 stations) diurnal variation of $PM_{2.5}$ for each period (JF, MAM, JJAS, and OND) in the revised MS. We have provided the information in the above comment#3.

*Comment#8 Bias correction methodology.*

*Although MERRA2 aerosol reanalysis products are better than non-assimilated products, it can have biases, therefore it was calibrated across India (spatially) and during2009-2017 (temporally) using the CPCB data measured at 80 sites mentioned in supplementary material table 1. To obtain the collocated CBCP and MERRA2 PM2.5,authors have either averaged all CPCB sites within a MERRA-2 grid (0.5◦×0.625◦)OR re-grid the MERRA-2 data from 0.5 x 0.625 degree resolution to 0.05 x 0.05 degree resolution and then extracted the PM2.5 values at CPCB coordinates (as per reply to the pre review comments). Authors use 50% CPCB data for bias correction and 50% for validation. Please clarify how the 50% data was selected, was it random or continuous.*

Response: We have modified the bias correction methodology in the revised MS, explained above in comment#3.

*First, authors need to address the issues related to CPCB data quality, its availability during 2009-2017 and its spatial representativeness as most of them are in Urban area and are within*

*the same grid. Second, the bias correction methodology needs further clarification as it seems as per the manuscript that authors do two types of bias correction (or calibration). One for in-situ 80 sites and another for the Indian grid. For80 sites, authors obtain a linear relation between MERRA2 and CPCB PM2.5 and then get the calibration factor as a function of CPCB PM2.5 which is then added to MERRA2 to correct it. (Line 164-171). For Indian grids, authors calculate the calibration factor as a function of MERRA2 2.5 value. To find out the linear regression, authors have binned the data in 500 bins (0-500 ug/m3) (in this way the data becomes independent of the time and location).*

Response:
First:
Apart from CPCB $PM_{2.5}$ dataset, we don't have any publicly available in-situ $PM_{2.5}$ observational dataset in terms of spatial and temporal resolution across the whole Indian region. So, we had to rely on the available CPCB urban $PM_{2.5}$ dataset for the validation part with the MERRA-2 dataset. As mentioned earlier, we took caution in handling CPCB data. We personally went through the entire raw data, and eliminate data from the period where it showed spurious and wild fluctuations. We have validated the CPCB data on the Indian state level as well (revised supplementary material). The state level data are generated from the given CPCB sites. In this case, the CPCB sites which are within the same grid can be considered as one state for example Delhi state which have 25 sites within one grid.

Regarding binning issue, we apologise for creating confusion on this issue. We did not bin the data in 500 bins. The bins actually represents the colour bins for generating the density plot (left panel Figure 1). We have modified the line regarding this issue in the revised MS.

*For the linear relation used for 80 sites, authors get a liner line with a slope of 0.228between CPCB and MERRA2. This shows that there is a huge underestimation of MERRA2 PM2.5 most probably because of the use of Urban PM2.5. It is even more surprising liner line between bias (CPCB-MERRA2) has a slope of 0.772. It can be interpreted that model bias has a better correlation then the model estimate. And if the model bias is more than the model estimate then one must rethink before using this data for further analysis.*

Response: The studies mentioned that lack of nitrate and underestimation of OC aerosol species in the GOCART module could influence the simulation of MERRA-2 $PM_{2.5}$ (Randles et al., 2016, Song et al., 2018), which may further responsible for the underestimation of MERRA-2 $PM_{2.5}$ data. So, the only way we have to correct the underestimation of MERRA-2 $PM_{2.5}$ is by use of some better bias correction methodology. Therefore, we have replaced the old bias correction method with the current percentile-based bias correction method, which provide a better result than the previous one.

*Finally, the authors find a bias-corrected relation BCM=0.99*CPCB+0.005. Rounding off and further simplification, this equation reduced to BCM=CPCB. It means, all theMERRA2 values are replaced by CPCB values. In this way, authors will certainly get good correlation (0.94) between bias-corrected MERRA2 and validation CPCB PM2.5.Authors can check and report the correlation between validation and the data used for bias correction. By using this methodology, you are overfitting the MEERA2 data. I don't think this is the right way to do the*

*bias correction. There are several papers on bias correction methodology that authors can refer to.*

Response: We thank to reviewer for pointing out this issue. We have modified our MERRA-2 PM$_{2.5}$ dataset and bias correction methodology according to the reviewer suggestion.

The validation on an hourly scale between reanalysis and in-situ data showed that there is a positive correlation (r = 0.57 with Npts > 400000) for the period of 2009-2017 (Figure 1). Indeed, we agree with the reviewer that the MERRA-2 data is highly biased w.r.t. the in-situ measurement and does not fully represent the rural sites, which could be considered as one of the limitations of the MERRA-2 dataset. However, based on the available large dataset points, we can argue that even without bias correction, both MERRA-2 and in-situ measurements are showing some significant positive correlation value (r = 0.57) which further signify that there is need to apply some bias correction method to improve the MERRA-2 dataset as 1:1 ratio w.r.t observational CPCB dataset.

Additionally, we would like to report that, modeling oriented studies either underestimate or overestimate the values with respect to the ground-based or space-based observational measurements. However, it is also true that models are also capable of capturing the more or less same spatial/temporal pattern with the observational data (in-situ and remote sensing). So, we would like to point out that the MERRA-2 dataset has also shown more or less same PM$_{2.5}$ diurnal pattern with in-situ CPCB over the Indian region.

We have introduced the information in the revised MS section 2.3 [Line 198-255] as:

**2.3 Calibration of MERRA-2 PM$_{2.5}$ with CPCB**

We calibrated hourly MERRA-2 PM$_{2.5}$ with coincident PM$_{2.5}$ data from 75 CPCB sites across the country for the period 2009-2017, as CPCB data are available only for these periods. The uncalibrated MERRA-2 PM$_{2.5}$ shows a correlation of 0.57 (significant at 95% CI) with coincident in-situ PM$_{2.5}$ (left panel of Figure 1). For bias correction, we used the percentile based bias correction methodology. We divided the MERRA-2 data at 10 percent interval and then calculated the relationship ($r = 0.9$) between median bias at every 10 percentile ranges between the two datasets (central panel of Figure 1). Then we adjusted MERRA-2 data with the calibration factors of the respective percentile ranges. Bias-corrected MERRA-2 at every grid (Figure S1) and median PM$_{2.5}$ at every 50 $\mu$g m$^{-3}$ interval (right panel of Figure 1) show improved correlation with the in-situ data. We note that MERRA-2 PM$_{2.5}$ is still underestimated at very high concentration (i.e. >300 $\mu$g m$^{-3}$); but since most of the country does not have any ground-based monitoring, we proceeded with our analysis with the calibrated MERRA-2 PM$_{2.5}$ to examine the diurnal pattern over India.

[Figure]

**Figure 1.** Scatter plot between (left) uncalibrated MERRA-2 and in-situ PM$_{2.5}$ data, (middle) median bias in MERRA-2 and in-situ PM$_{2.5}$ at every 10 percentile ranges, and (right) calibrated median PM$_{2.5}$ from MERRA-2 and in-situ at every 50 $\mu$g m$^{-3}$ interval. Spatial and temporal matching is done by averaging data from all ground-based monitoring sites falling within a single MERRA-2 grid for 1-hr duration.

**2.4 Diurnal patterns in calibrated MERRA-2 and CPCB data**

We have further analysed the diurnal variation of PM$_{2.5}$ in each season (JF, MAM, JJAS and OND) from calibrated MERRA-2 and CPCB data (Figure 2). The black, blue and red lines represent the mean of all 75 stations (2009-2017) from CPCB, uncalibrated MERRA-2, and calibrated MERRA-2, respectively.

[Figure]

**Figure 2**. Diurnal variation of PM$_{2.5}$ of CPCB, uncalibrated MERRA-2 (without bias correction, WBC) and calibrated MERRA-2 (after correcting bias, ABC) for OND, JF, MAM, and JJAS period. The shaded regions represent ±1σ around seasonal mean across India.

Two observations are notable. Firstly, calibrated MERRA-2 PM$_{2.5}$ is much closer to CPCB data relative to the uncalibrated MERRA-2 data across 24-hr period. Secondly, the calibrated MERRA-2 PM$_{2.5}$ is able to mimic the observed diurnal pattern in all the seasons except the winter (JF) when the low bias of calibrated MERRA-2 PM$_{2.5}$ is prominent. Capturing extreme pollution level and very shallow boundary layer is always challenging for the models. MERRA-2 captures the variation of high (i.e. early morning or late evening) and low PM$_{2.5}$ concentration (noontime to 4 p.m.).

We also observed the same pattern for different states (Supp. R1-R3) which covers the north, west, south, and east part of India. We observed the high (i.e. early morning or late evening) and low (noontime to 4 p.m.) pollution during winters and pre-monsoon/monsoon respectively for each state.

The comparison suggests that there is further need for an improvement in the simulation of planetary boundary height in the GEOS-5 model. Other likely factors for the observed low bias could be lack of nitrate and underestimation of OC aerosol species in the GOCART module (Randles et al., 2016, Song et al., 2018). According to Rienecker et al. (2011), two types of schemes are introduced in the GEOS-5 model for simulating the atmospheric boundary layer in the MERRA-2 reanalysis dataset. The first scheme (Louis et al., 1982) is based on the planetary stable condition in which no planetary boundary layer of clouds involved. The second scheme (Lock et al., 2000) is based on the unstable or cloud-topped planetary boundary layer condition involved. Additionally, the GEOS-5 model uses two more schemes based on orographic conditions such as orographic gravity wave drag (McFarlane 1987) and non-orographic waves (Garcia and Boville 1994).

*Comment#9 Overall comment*
*I have no further comments on the rest of the analysis as it depends on how good is the bias-corrected MERRA2 data. As the moments it appears that 1. MERRA2 PM2.5 may not be suitable for hourly analysis. 2. MARRA2 PM2.5 cannot be used for trend analysis because of constant emissions after 2008. 3. MERRA2 aerosols are not suitable for Urban PM2.5 analysis. Because of this, The MERRA2model bias is more than the MERRA2 estimates. This suggests urban PM2.5 should not be used for bias correction. 4. A robust method of bias correction is required.*

1. We summarize that MeRRA-2 $PM_{2.5}$ mimics high and low $PM_{2.5}$ concentration around the same time scale (high during morning/late evening and low during noon time to 4pm) as we observed in CPCB $PM_{2.5}$ dataset reasonably well. Post bias correction, the MERRA-2 data is much closer to the in-situ data at every hour. Hence we believe that MERRA-2 $PM_{2.5}$ are suitable for hourly analysis. We discussed the limitations in the revised MS.

2. As we have mentioned in comment#1 that MERRA-2 datasets are also using 4 more different sensors (i.e. MODIS-Terra, Aqua, MISR, AVHRR, and AERONET) along with the emission inventories. So, the years those are absent during the emission inventories period, satellite and in-situ datasets have been used to simulate the aerosol products and these 4 sensors are highly reliable and acceptable for global and different regional studies. Therefore, we rely on the use of the MERRA-2 dataset to drive the trend over the Indian region.

3. As we mentioned in the comment#6 that apart from CPCB $PM_{2.5}$ dataset, we don't have any publicly available in-situ $PM_{2.5}$ observational dataset in terms of spatial and temporal resolution across the whole Indian region. So, we had to rely on the available CPCB urban $PM_{2.5}$ dataset for the validation part with the MERRA-2 dataset. We took

caution in handling CPCB data. We personally went through the entire raw data, and eliminate data from the period where it showed spurious and wild fluctuations. We understand the importance of data quality by the regulatory agencies, but we cannot do more than this. WHO and exposure data used in GBD (Shaddick et al., 2018) also had to calibrate with CPCB network. Firstly, calibrated MERRA-2 $PM_{2.5}$ is much closer to CPCB data relative to the uncalibrated MERRA-2 data across 24-hr period. Secondly, the calibrated MERRA-2 $PM_{2.5}$ is able to mimic the observed diurnal pattern in all the seasons except the winter (JF) when the low bias of calibrated MERRA-2 $PM_{2.5}$ is prominent. Capturing extreme pollution level and very shallow boundary layer is always challenging for the models. MERRA-2 captures the variation of high (i.e. early morning or late evening) and low $PM_{2.5}$ concentration (noontime to 4 p.m.). We also observed the same pattern for different states (Supp. R1-R3) which covers the north, west, south, and east part of India. We observed the high and low pollution during winters and pre-monsoon/monsoon respectively for each state respectively.

4.  We have revised the bias correction method as per requested by the reviewers. The detailed information of the bias correction and validation part is given in the above comment#8 (sub-comment #4) and in the revise MS [Line 198-255].

*Hence, I am less confident that the results presented in the paper are valid. This is important because the authors claim that these results will help formulate better air pollution mitigation plans by evidence-based policy actions at the regional and national levels. I feel that authors need to be extra cautious and discuss the limitations before publishing these kinds of results. At the moment, it would be appropriate the perform a detailed evaluation of the MERRA2 products over India and present the biases and uncertainties across different temporal scales and geographical regions of India.*

Response: Any exposure modelling has uncertainty. $PM_{2.5}$ data used in GBD or by WHO for burden estimation, $PM_{2.5}$ was derived by satellite AOD using a model and then calibrated by CPCB data. We agree that MERRA-2 data also have uncertainty but as we have demonstrated that the data are much closer to in-situ data at every hour post calibration. India lacks a robust and dense monitoring network and suffers terribly from air pollution. Therefore, it is important to generate data using hybrid approach. As discussed earlier, we modified $PM_{2.5}$ estimation method and bias correction method. MERRA-2 calibrated $PM_{2.5}$ still has low bias especially at very high $PM_{2.5}$ concentration but they mimic the diurnal pattern quite reasonably. However our main motivation was to examine the diurnal amplitude of $PM_{2.5}$ exposure in India over long period of time. We believe the calibrated $PM_{2.5}$ data may not be perfect but they are good enough to capture the broad spatial and temporal variation at hourly time scale. Few model based studies demonstrated large diurnal variation of $PM_{2.5}$ over India but they are limited to shorter duration and plagued by static emission inventory. The diurnal pattern has not been examined before at a longer time period. Even within the uncertainty, we could demonstrate the diurnal variation in terms of meteorology (variation in PBL height and rainfall), and therefore, our study has important contributions. In future, as the modelling techniques will improve, the uncertainty in such hybrid approach will reduce.

*Comment#10 Some of the minor suggestions*

*Use either bias corrected MERRA (BCM) or calibrated uniformly. This paper has not been referred: Central Pollution Control Board (CPCB) Ambient air quality statistics for Indian metro cities, Central Pollution Control Board, Zonal Office, Bangalore, 2003. Authors can discuss India specific assimilation used for MERRA2 in detail as indicated by referee.*

*Response: Done*